# Normative approaches to neural coding and behavior

**Ann M. Hermundstad**

Janelia Research Campus, Howard Hughes Medical Institute

hermundstada@janelia.hhmi.org

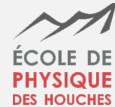

*Part of the 2023-07: Theoretical Biological Physics 2023 collection*
*Session 121 of the Les Houches School, July 2023*
*published in the Les Houches Summer School Lecture Notes series*

## Abstract

These are a brief set of lectures notes for lectures given at the Les Houches Summer School in Theoretical Biological Physics in July 2023. In these notes, I provide an introduction to some of the theoretical frameworks that are used to understand how the brain makes sense of incoming signals from the environment to ultimately guide effective behavior. I then discuss how we can apply these frameworks to understand the structure and function of real brains.

| Received | 22-01-2024 |
| Accepted | 18-04-2024 |
| Published | 13-08-2024 |

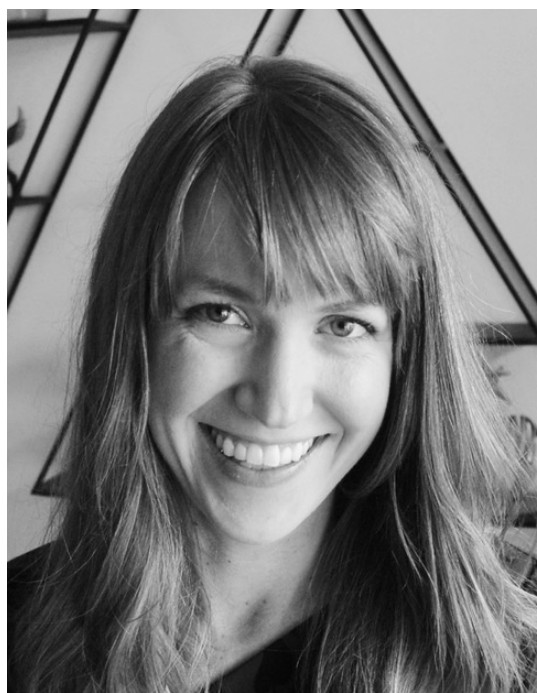

## 1 Introduction

The brain is responsible for a myriad of different computations that enable animals to behave effectively in an ever-changing world. The brain must sense signals in the environment, make inferences and predictions about the causes and consequences of those signals, and must plan and execute appropriate behavior based on those inferences and predictions.

One way that we try to test our understanding of such computations is by building artificial systems that can mimic the computations and behaviors of real animals. To date, our best artificial systems pale in comparison to the capabilities of real brains (but large language models provide an interesting counterpoint); real brains continue to outperform our best artificial systems. And yet, real brains have many limitations that artificial systems do not—limitations

in speed and precision, and in energy use and storage capacity, for example. In order to overcome these limitations, it is thought that brains have evolved smart strategies to exploit the lawfulness of the world in order to achieve "good enough" performance. This can happen over many different timescales: over the course of evolution, development, learning, and adaptation. This idea necessarily implies that there are better and worse solutions for exploiting the lawfulness of the world to secure a performance advantage. One goal of theoretical neuroscience is to understand the space of such solutions—how many there might be, their advantages and disadvantages, and, ultimately, why biology might have persisted with one solution over another. This involves understanding the possible functions of, and constraints faced by, different biological systems.

In the following notes, I broadly organize these computations beginning with sensory input and moving toward motor output. In between, the brain has to transform sensory input into appropriate inferences, decisions, and plans of action. In the first section, I begin by discussing some theoretical frameworks that are used to understand computation across animals and brain regions. In the second section, I discuss how we can use these frameworks to try to close the loop in a single system.

## 1.1 The normative approach

Before diving in, it's worth briefly discussing the philosophical and methodological approach that we will take throughout these discussions. Much of theoretical and computational neuroscience concerns itself with how the brain works: if I can give you a mathematical model that accurately reproduces an experimental finding and makes testable predictions for new experiments, I have understood something about how a certain biological process might unfold. A smaller swath of theoretical neuroscience concerns itself with understanding not just *how* a process might unfold, by *why* it unfolds in this particular way. Why, of all of the possible strategies that the brain could have adopted to solve a particular problem, is *this* the strategy it adopted? There are many possible answers to this question, some of which could be 'this is just what evolution stumbled upon'. But in many cases, the putative answer to this question helps us understand how a given strategy might be advantageous with respect to performing a particular function, in a particular setting, and subject to a particular set of biological or physical constraints.

To give a more concrete example: consider your own visual perception of the world. This perception is something that can be measured and tested by asking you to discriminate different visual stimuli. By designing those stimuli in a way that isolates different statistical properties of the world, it is possible to examine which properties of the visual world are easy for you to see, and which are more difficult. This is the purview of the field of psychophysics.

For example, if you are presented with visual patterns hidden in a noisy background, you could be asked to locate the pattern in the noise. The more accurately you can do this, the better you are able to discriminate these patterns from random noise. It is possible to design these visual patterns with certain statistical structure by enforcing certain multi-point correlations in light intensity between nearby pixels [1, 2]. Now, if we pick a particular order of correlation—say, 4th order—it turns out that people can easily discriminate some 4th order patterns, but struggle to discriminate others, even though they have the same amount of statistical structure [3]. This is an empirical observation, and we could use this to build a model that can reproduce this finding. But *why* it is that we should be better able to see some patterns over others? Why those particular patterns?

The answer is that the patterns that we can easily discriminate are those patterns that are most informative about our natural visual world [3, 4]. Were our visual world organized in some different way, this would imply that we would be good at seeing different sets of patterns. Just by knowing the statistical structure of the visual world, we can predict how sensitive

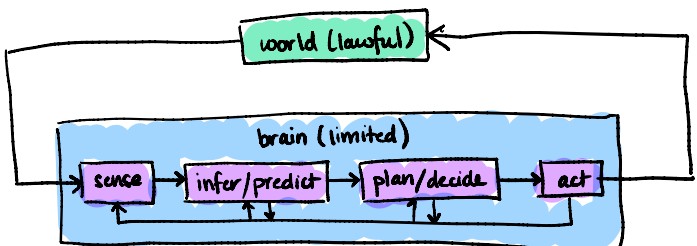

Figure 1: The brain has to use its limited resources to make sense of signals in the environment and guide appropriate actions.

a person will be to different types of visual patterns in the world, and we can do this in a parameter-free manner. This same idea—that we should be most sensitive to sensory signals that are most informative about our surroundings—has been used to explain a wide range of neural and behavioral responses across different species and different sensory systems.

Implicit in this statement, however, is the fact that our sensory systems are constrained: we have to prioritize some sensory signals, some visual patterns, over others. If we had infinite resources, we would not need to prioritize—we would in principle be able to fully discriminate any visual pattern from any other. So a more precise version of the previous statement is that our sensory systems should use minimal resources to maximize the information that they convey about the natural world in which we operate.

This is an example of a 'normative' statement: it postulates a goal or a function that a system is trying to achieve (potentially at the expense of other goals). The mere fact that we can propose and test normative statements reflects that fact that the world around us obeys predictable laws that can be exploited by biological systems, and that these systems themselves are limited in their ability to exploit these laws. If biological systems had infinite memory of all past states of the world and could perfectly predict and access all future states, there would be no need to prioritize some goals over others. Similarly, if the world were completely random and unstructured, there would be nothing to prioritize. In this way, normative theories relate the lawfulness of the world to the performance advantage that can be achieved by exploiting that lawfulness. In formulating a normative theory, we postulate (i) a function to be performed, (ii) a context in which that function will be performed, and (iii) a set of constraints on the system that performs the given function in the given context. We then derive the optimal solution for achieving this particular function in this context and subject to these constraints. In the example above about our own visual sensitivity to different patterns, (i) the function was to maximize information about incoming visual signals, (ii) the context was the natural visual world, and (iii) the constraints were limitations in bandwidth (i.e., how many signals could be reliably transmitted downstream).

In what follows, we will use this normative approach to consider how sensory systems might optimally encode and infer features of the external environment, and how downstream brain regions can in turn guide optimal behavior based on these inferences (Figure 1).

## 2   Frameworks that bridge systems

In this first section, we will briefly discuss three different normative theoretical frameworks—efficient coding (Section 2.1), Bayesian inference (Section 2.2), and reinforcement learning (Section 2.3)—that have been used to understand the computational processes involved in sensory coding, inference, and action selection.

## 2.1 Sensory coding

The world around us is full of different signals—patterns of light, sounds, smells—that we use to make sense of our surroundings. At the very earliest stages of processing, the brain must encode these external signals in internal ones. These internal signals are then used to build our perceptions of the world, and make decisions based on them. It is remarkable that our entire understanding of the world is created internally and is built from signals that are first transduced in our peripheral nervous system.

From the perspective of the brain, external signals in the world are not directly accessible, and can at best be disambiguated from the responses of neurons. This places limitations on the accuracy with which the brain can "know" the actual state of the environment. To examine the process of disambiguation, let's take a brief detour to consider the following example (taken from David MacKay's book [5]), which might already be familiar to many readers: you are given 12 balls that look identical, but one ball is heavier or lighter than the rest. You are given one scale that can compare any two sets of balls. How do you determine which ball is the odd ball, and whether it is heavier or lighter, in as few uses of the scale as possible? If you haven't already solved this problem before, it's worth sketching out your solution before moving on.

Stated another way, the goal is to design a set of measurements to disambiguate a set of hypotheses about the world as efficiently as possible (Figure 2). Here, there are 24 hypotheses in total: there are 12 balls, and each ball could be heavier or lighter, for a total of 24 possibilities. Each measurement corresponds to a usage of the scale, and there are 3 possible outcomes per measurement: the lefthand side of the scale is heavier, lighter, or equal to the righthand side. Thus, given 2 measurements, it is possible to disambiguate $3^2 = 9$ hypotheses; given 3 measurements, it is possible to disambiguate $3^3 = 27$ hypotheses. From this, we know that it is possible to disambiguate all 24 of our hypotheses with a total of 3 measurements. The next step is to design those measurements.

The most efficient way to design a measurement is to maximize the entropy over the set of possible outcomes for that measurement. Here, by "most efficient", we mean that the measurement will reduce our uncertainty as much as possible, given the resolution of the measurement device. This is an example of a normative statement: it posits an optimal solution (maximizing the entropy over possible measurement outcomes) for a particular problem (disambiguating a fixed set of hypotheses as quickly as possible) and subject to a particular set of constraints (subject to the resolution of the measurement device).

For example, if we begin with 24 hypotheses and have 3 possible outcomes, we want to assign $24/3 = 8$ hypotheses to each outcome. One way to do this would be to divide our 12 of balls into 3 sets: balls 1-4 (set 1), balls 5-8 (set 2), and balls 9-12 (set 3). We can then compare any two sets (e.g. sets 1 and 2). There are then three possible outcomes that evenly break apart our hypothesis space:

1. **Set 1 is heavier.** There are then 8 remaining hypotheses: set 1 has the heavier ball (4 hypotheses), or set 2 has the lighter ball (4 hypotheses)

2. **Set 1 is lighter.** There are again 8 remaining hypotheses: set 1 has the lighter ball, or set 2 has the heavier ball

3. **Both sets are equal.** Again, 8 remaining hypotheses: set 3 has the heavier ball, or set 3 has the lighter ball

I will leave it up to the reader to carry this logic through to design the remaining two sets of measurements. When applying this measurement technique in succession, we can identify the odd ball, and determine whether it is heavier or lighter than the rest, in the fewest possible measurements (it's easy to verify that a different measurement scheme would require more

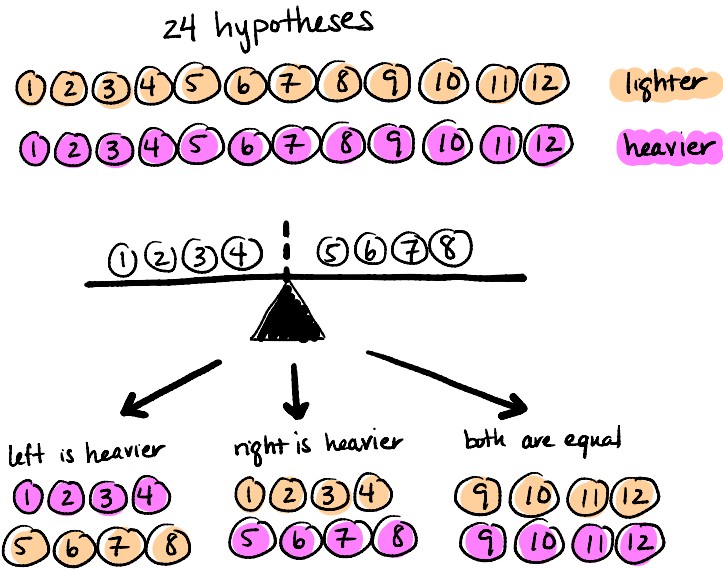

Figure 2: Designing measurements to disambiguate hypotheses (adapted from [5]).

than 3 measurements on average). This is precisely because we reduce as much as possible our uncertainty about the unknown input—and gain as much information as possible about that input—with the outcome of each successive measurement. However, had we been given only a single measurement, we would not have been able to fully disambiguate the identity and relative weight of the odd ball. For example, if we performed the single measurement shown in Figure 2 and observed that the left side of the scale was heavier, we would at best be able to say that one of balls 1-4 was heavier or one of balls 5-8 was lighter. This highlights that our ability to disambiguate is limited by the resolution of our measurement device.

We can apply a similar perspective to the process of disambiguating signals in the brain. For example, consider a neuron that acts as a device to measure incoming light signals. Given some knowledge about the distribution of light signals that could be present in the environment, one can ask how to design the response function of the neuron to best disambiguate these signals subject to the limitations of the device itself. If the neuron can produce a binary output (e.g., it can spike or not spike), then this optimal response function should partition the incoming distribution of light signals into two equal probability chunks, and assign one output to each chunk. This would, for example, lead to a scenario in which the neuron spikes if the incoming light signal is greater than the median of the distribution, and is silent if the incoming signal is less than the median. By designing a response function that partitions the incoming distribution into equal probability chunks, we are guaranteed that the entropy of the neural response will be maximized given the distribution of input signals. This is often referred to as 'histogram equalization', because the histogram of neural responses will be flat. However, it's important to note that there are many ways to partition the distribution into equal probability chunks, and thus we often have to invoke other constraints (such as continuity of the response function) before comparing to biology.

This idea forms the basis of one of the most influential normative frameworks in neuroscience: efficient coding. Efficient coding posits that sensory systems maximize the information that they convey to downstream brain regions about incoming sensory signals, and in doing so exploit the statistics of the environment in which an organism must function [6]. This hypothesis was first formulated by Fred Attneave [7] and Horace Barlow [8]. Given an input message (here, our stimulus $S$) and an output message (here, our neural response $R$),

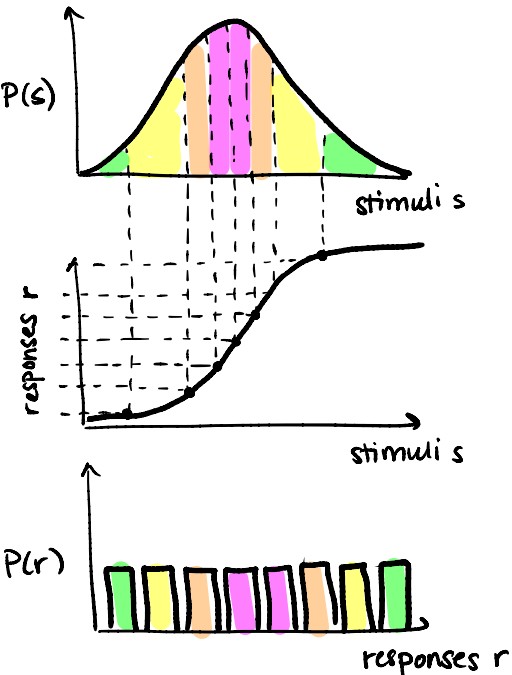

Figure 3: Maximizing entropy about the distribution of incoming stimuli (adapted from [9]).

Barlow hypothesized that the goal of the early nervous system is to maximize the information that the output message conveys about the input message. The mutual information between the input and output is given by:

$$I(R;S) = H(R) - H(R|S), \tag{1}$$

where $H(R)$ is the entropy of the response, and $H(R|S)$ is the conditional entropy of the response given the stimulus (see Appendix A for derivation). In the limit of low input noise (i.e., the stimulus is reliable), $H(R|S)$ is close to zero, and maximizing mutual information is equivalent to maximizing the entropy of the neural response. Barlow used this idea to define a quantity called the 'redundancy', $\mathcal{R}$, to be minimized:

$$\mathcal{R} = 1 - H(R)/C, \tag{2}$$

where $C$ is the capacity of the channel along which the messages are sent. It's worth noting that entropy maximization, or information maximization more broadly, does not concern itself with how this information will be used downstream. In fact, Attneave discussed this as one of the advantages of the efficient coding framework—that it could be used to understand the first stages of sensory processing, without having to know the putative relevance of different sensory signals [7].

More broadly, this idea can be used to derive the entropy-maximizing set of neural responses subject to different constraints on the response distribution (Appendix B). For example, given a constraint on the total number $N$ of discriminable responses that a neuron can produce, the entropy-maximizing response distribution is flat (i.e., $P(r) = 1/N$; this is our histogram equalization). Given a constraint on the mean firing rate $\mu$ of the neuron, the response distribution is exponential: $P(r) = \exp(-r/\mu)/\mu$. And given a constraint on the variance in firing rate $\sigma^2$, the response distribution is Gaussian: $P(r) = \exp(-(r-\mu)^2/2\sigma^2)/\sqrt{2\pi\sigma^2}$.

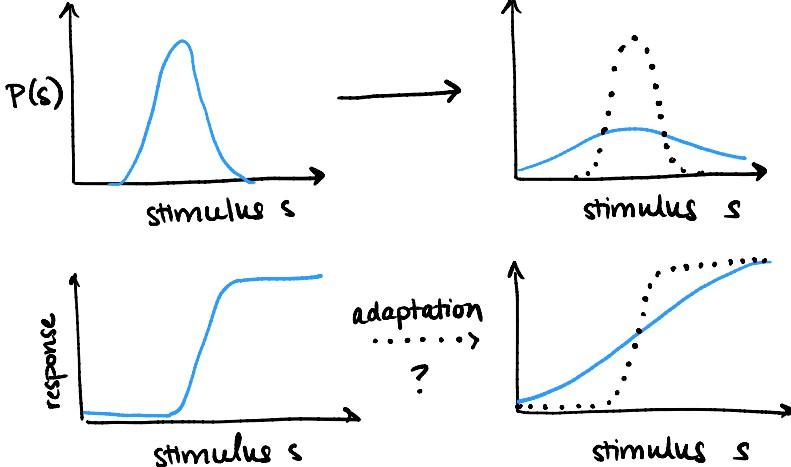

Figure 4: If the statistics of incoming sensory signals change over time, neural responses should adapt to the change.

Barlow's hypothesis was first tested experimentally in 1981 by Simon Laughlin [9], who showed that he could predict the responses of motion-sensitive neurons in the visual system of the blowfly using only the distribution of light signals found in the natural environment (Figure 3). Since then, there have been numerous studies that have sought to understand different aspects of sensory processing in terms of maximizing information about the distribution of inputs that a sensory system encounters in its environment, or in terms of minimizing error in the reconstruction of those inputs [6]. The optimal sensory coding scheme depends on the specific statistics of the environment, the resource limitations and sources of noise within the encoding scheme, and the specific objective function used to optimize the encoding scheme. As a result, there is no single characterization of an efficient code. For example, in the limit of low input noise, when incoming sensory signals are reliable, optimal coding strategies tend to decorrelate these signals (consistent with Barlow's redundancy reduction hypothesis). In the limit of high input noise, optimal coding strategies tend to average incoming signals in order to combat the adversarial effects of noise. These different coding regimes emerge from the same underlying framework but under different assumptions, and are predicted at the level of single cells [10–13], population codes [14,15], and behavior [4].

A key assumption underlying all of this work is that sensory systems have evolved to exploit the particular statistics of the environment in which an organism must function. If that environment were to change, we would expect sensory systems to change as well—a point that we will examine in the next section.

## 2.2 Inference

*This section covers material from Młynarski & Hermundstad (2018) and (2021).*

In the previous section, we discussed how sensory systems could have evolved to exploit the statistics of the environment. This implies that if the environment were to change, sensory systems should change with it (Figure 4). In this way, efficient coding has provided a normative perspective on sensory adaptation [16,17].

However, the statistics that are most relevant for adapting to a change in the environment are not necessarily the same as those that should be optimally encoded in steady state [18]. Moreover, the different sensory signals carry different relevance for downstream computations,

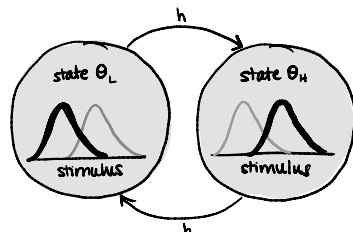

Figure 5: Example nonstationary environment in which the mean (or variance) of a Gaussian stimulus distribution switches between a low and high value over time.

which can necessitate different coding strategies [19]. In this section, we treat both of these questions.

In the previous section, we considered how to best design a neural response function in order to efficiently encode signals $s$ drawn from a distribution $P(s)$. We will now assume that this distribution can be parameterized by a latent variable $\theta$, such that we can specify $P(s|\theta)$ (we will also refer to $\theta$ as the "state" of the environment). For example, for a Gaussian stimulus distribution, $\theta$ can paramterize either the mean or the variance of the distribution. We can then compare a neural response function that maximizes the information about incoming sensory stimuli (as discussed in the previous section) with a response function that maximizes information about the underlying latent parameters of the distribution. In other words, when designing the optimal response function, we can concern ourselves with disambiguating the current sensory stimulus $s$ or the underlying latent state $\theta$.

The parameter $\theta$ is not observable, and thus its value must be inferred from incoming sensory signals. Such a computations falls within the purview of Bayesian inference, which specifies the statistically-optimal computation for inferring a latent variable from a set of measurements. We will consider a simple case described above, in which $\theta$ specifies either the mean or the variance of a Gaussian distribution. We will also assume that $\theta$ takes on one of two values—a high value or a low value. Finally, we will assume that the environment can dynamically switch between these two different values with a small but fixed probability $h$ per timestep (Figure 5). At each timestep $t$, a signal $s_t \sim p(\cdot|\theta_t)$ is sampled from a distribution parameterized by $\theta_t$. This setup mimics one that is widely used in sensory neuroscience to study the dynamics of adaptation to changes in the mean and variance of sensory stimuli [16,17,20–23]. The goal of the inference, then, is to use the history of past sensory signals $s_{\tau \leq t}$ determine whether $\theta_t$ is high ($\theta_t = \theta^H$) or low ($\theta_t = \theta^L$) at a particular timestep.

To illustrate the inference process, we will derive an "ideal observer" that has certain knowledge about the environment, and can use this knowledge to optimally perform the inference. In this setting, we will assume that the observer knows (1) that the environment exists in one of two states (high or low); (2) the identity of the two states (i.e., the values $\theta^H$ and $\theta^L$); (3) the switching probability $h$; (4) the form of the stimulus distribution; and (5) the values of all remaining parameters of the stimulus distribution (i.e., if $\theta$ parameterizes the mean of the Gaussian distribution, we assume that the ideal observer knows the variance of that distribution; if $\theta$ parameterizes the variance, we assume that the ideal observer knows the mean). In other words, the observer is armed with a great deal of prior knowledge; the only thing that the observer does not know is the current value of $\theta_t$, and the specific signal that will be sampled from $p(s|\theta_t)$. To infer $\theta_t$, the ideal observer can use the history of observed signals $s_{\tau \leq t}$ to construct the posterior distribution $P(\theta_t|s_{\tau \leq t})$ (also called the "posterior belief", or "belief distribution"):

$$P(\theta_t|s_{\tau \leq t}) = \frac{1}{\Omega} P(s_t|\theta_t) \sum_{\theta_{t-1}} P(\theta_t|\theta_{t-1}) P(\theta_{t-1}|s_{\tau \leq t-1}), \tag{3}$$

where

$$P(\theta_t|\theta_{t-1}) = \begin{cases} (1-h), & \theta_t = \theta_{t-1}, \\ h, & \theta_t \neq \theta_{t-1}, \end{cases} \tag{4}$$

specifies the probability that the environment switched states at time $t$, $P(s_t|\theta_t)$ is the stimulus distribution, and $\Omega$ is a normalization constant (see Appendix C for a derivation of this distribution). For an environment that consists of only two states ($\theta^L$ and $\theta^H$), the posterior distribution is defined over two values, and can thus be specified by a single number $P_t^L \equiv P(\theta_t = \theta^L|s_{\tau \leq t})$ that specifies the probability that the environment is in the low state (and we can use this to compute $P_t^H = (1 - P_t^L)$). We can use this to rewrite the posterior as:

$$P_t^L = \frac{1}{\Omega} P(s_t|\theta_t = \theta^L)\big[(1-h)P_{t-1}^L + h(1-P_{t-1}^L)\big]. \tag{5}$$

From this expression, one can see that the posterior belief that the environment was in the low state at time $t$ depends on the prior belief at time $t - 1$, weighted by the probability that the environment stayed in the low state, and by the likelihood of sampling the observed signal $s_t$ in the low state.

We can use this belief distribution to construct a point estimate $\hat{\theta}_t$ of the current state $\theta_t$, for example by computing the mean of the posterior:

$$\begin{aligned} \hat{\theta}_t &= \langle P(\theta_t|s_{\tau \leq t})\rangle_{\theta_t} \\ &= \theta^L P_t^L + \theta^H(1 - P_t^L). \end{aligned} \tag{6}$$

We choose this, rather than the maximum a posteriori probability (MAP) estimate, because this is the optimal point estimate that minimizes the mean-squared error between $\theta_t$ and $\hat{\theta}_t$ [24]. More generally, in higher dimensional settings, a point estimate serves as a compact summary of the full posterior distribution.

Standard Bayesian inference assumes that the sensory signal $s_t$ is directly used to update the posterior belief $P(\theta_t|s_{\tau \leq t})$. However, as discussed in the previous section, any incoming sensory signals must be encoded in neural responses before they are used to perform any downstream computations. Because neurons have finite precision and bandwidth constraints, there is necessarily loss in this encoding step. As a result, the choice of encoding schemes will impact any downstream inferences (Figure 6).

To mitigate the negative impact of the encoding, we can design an encoding code scheme that preserves information about the incoming stimuli that is relevant for updating the posterior belief. For example, consider an encoder with a response function of the form:

$$r_t(s_t; k, s_0) = \frac{1}{1 + \exp(-k(s_t - s_0))} + \eta, \tag{7}$$

where the parameters $k$ and $s_0$ respectively control the slope and offset of this sigmoidal function, and $\eta$ is additive noise (alternatively, we can discretize the output $r_t$ into a set of $N$ discriminable response levels, analogous to the encoding scheme discussed in Section 2.1). This response function has finite resolution that it can devote to incoming signals; because of the saturating nature of this response function, signals that are sufficiently large or small will not be distinguishable from one another. The output of this response function, $r_t$, can then be decoded to construct an estimate $\hat{s}_t$ of the incoming sensory signal. For the purposes of this discussion, we will assume that we can construct and optimize a simple linear decoder to get the estimate $\hat{s}_t$. The Bayesian observer must then construct a posterior belief $P(\theta_t|\hat{s}_{\tau \leq t})$ built from the history of past signal *estimates*, rather than directly using the history of true signal values.

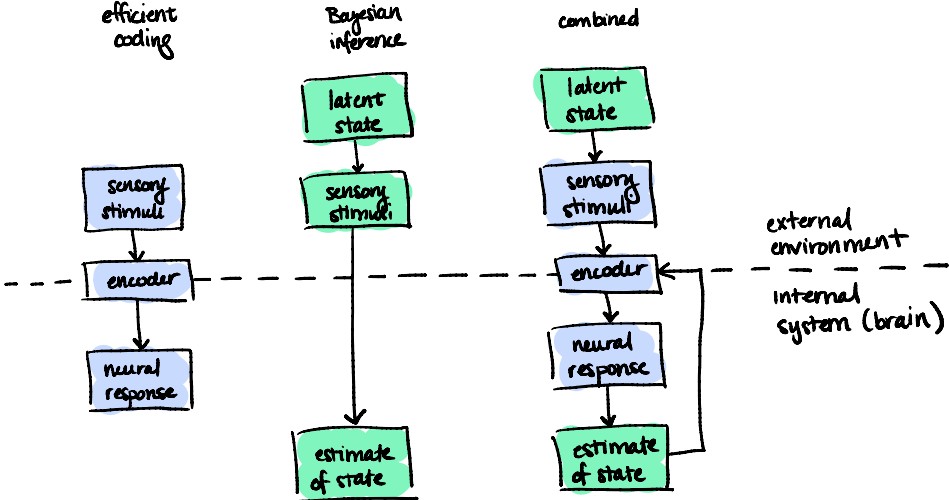

Figure 6: Different architectures of efficient coding and Bayesian inference.

One advantage of choosing a parameterized encoder is that we can directly optimize the parameters $k$ and $s_0$ of the encoder (as well as the parameters of the linear decoder). For the sake of illustration, we will consider two different objective functions for optimizing these parameters:

$$\operatorname{argmin}_{k,s_0} \left\langle (\hat{s} - s)^2 \right\rangle_{P(s_t|\hat{\theta}_t)} \tag{8}$$

and

$$\operatorname{argmin}_{k,s_0} \left\langle \left( \hat{\theta}(s) - \hat{\theta}(\hat{s}) \right)^2 \right\rangle_{P(s_t|\hat{\theta}_t)}. \tag{9}$$

The first of these objective functions prioritizes an accurate estimate of incoming sensory signals by minimizing the "reconstruction error"; the second of these objectives prioritizes an accurate estimate of the latent state of the distribution from which the signals were generated by minimizing the "inference error". In contrast to typical formulations of efficient coding, which optimize the encoder based on the true distribution of incoming stimuli $P(s_t|\theta_t)$, both of these objective functions optimize the encoder based on the *current belief* about the incoming stimulus distribution, $P(s_t|\hat{\theta}_t)$ [18, 19]. As a result, any inaccuracies in this belief will shape the current encoding scheme, and any inaccuracies in encoding will in turn shape the evolution of the posterior belief. This can be viewed as a form of adaptive coding, where the output of the encoder is used to update the posterior belief, and the posterior belief is used, in turn, to adapt the encoding on the next timestep. Moreover, rather than using the current estimate $\hat{\theta}_t$ to update the encoder, a better strategy is to use a prediction $\vec{\theta}_{t+1}$ about the state of the environment at the next timestep, and optimize the encoder based on the distribution $P(s_t|\vec{\theta}_{t+1})$ (note that here and in the following paragraph, we use vectors to denote point predictions at time $t+1$). The point prediction $\vec{\theta}_{t+1}$ can be obtained analogously to the point estimate $\hat{\theta}_t$, using the predicted posterior:

$$P(\theta_{t+1}|s_{\tau \leq t}) = \sum_{\theta_t} P(\theta_{t+1}|\theta_t) P(\theta_t|s_{\tau \leq t}). \tag{10}$$

As should be expected, the two objective functions in Eqs. 8-9 lead to different optimal parameters depending on the current prediction $\vec{\theta}_{t+1}$. Since this prediction is varying in time, the optimal encoding parameters will also vary in time (Figure 7). This can easily be seen

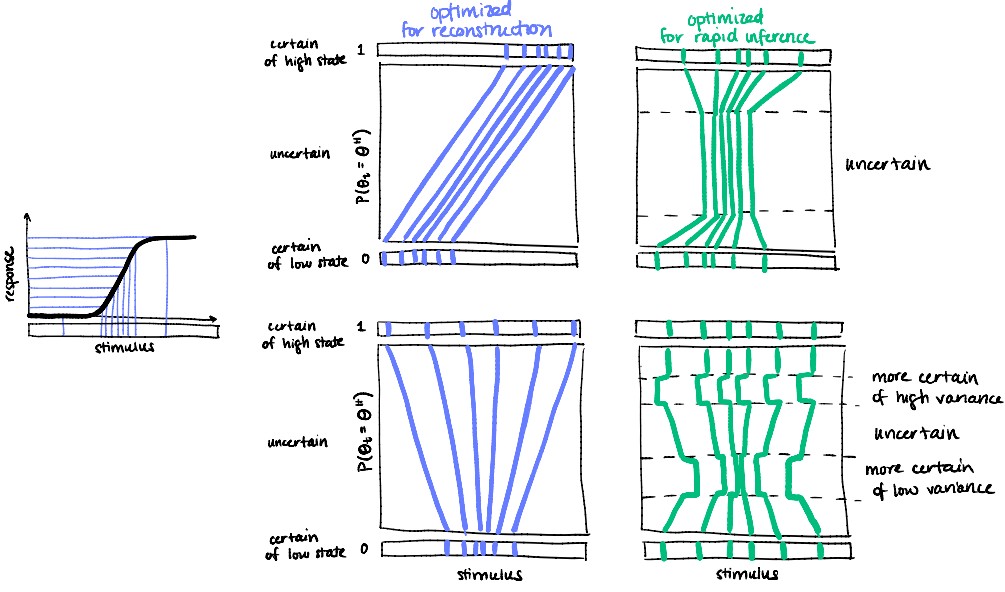

Figure 7: Optimal tuning curves (far left schematic) adapt based on uncertainty, shown for mean adaptation (top row) and variance adaptation (bottom row). Adapted from [18].

through numerical optimization. For example, the first objective function leads to an encoder that tracks the current prediction of the incoming stimulus distribution: the optimal offset tracks the predicted mean of the stimulus distribution, $s_0^* = \vec{\mu}_{t+1}$, and the optimal slope tracks the inverse of the predicted standard deviation of the stimulus distribution, $k^* = 1/\vec{\sigma}_{t+1}$. As a result, these parameters vary continuously with the current posterior belief (blue curves in Figure 7). In the limit that the observer is certain about the current state of the environment and maintains a correct estimate of that state, the encoder is aligned with the true incoming stimulus distribution, and prioritizes accurate encoding of stimuli from that distribution, analogous to classic efficient coding. However, when the state of the environment changes, the current encoder is misaligned with the new distribution, and can thus be slow to detect and adapt to a change. As a result, when averaged over time, this strategy often suffers from higher reconstruction error than a strategy that is optimized for inference. This highlights the fact that an encoding that performs optimal reconstruction of sensory signals does not necessarily support optimal adaptation to changes in the distribution of those signals.

In contrast, the second objective function is optimized for inference and leads to an encoder whose parameters vary discontinuously with the current prediction of the incoming stimulus distribution (green curves in Figure 7). The resulting encoder performs a form of uncertainty-dependent change detection: when the observer is certain about the current state of the environment (and maintains a correct estimate of that state), the encoder shifts its finite resolution away from the true stimulus distribution, in anticipation of changes that might occur in the future. When the state of the environment changes and the observer becomes more uncertain, the encoder takes a form that can best discriminate that change: for mean estimation, the optimal encoding function is sharp and centered between the two candidate distributions; for variance estimation, the optimal encoding function is shifted toward the tail of the high variance distribution (if more certain of the high variance state) or centered about the low variance distribution (if more certain of the low variance state). As a result, this encoder is less accurate in steady state, but more sensitive and faster to adapt to changes, and can achieve lower average reconstruction error than a strategy that is optimized for reconstruction.

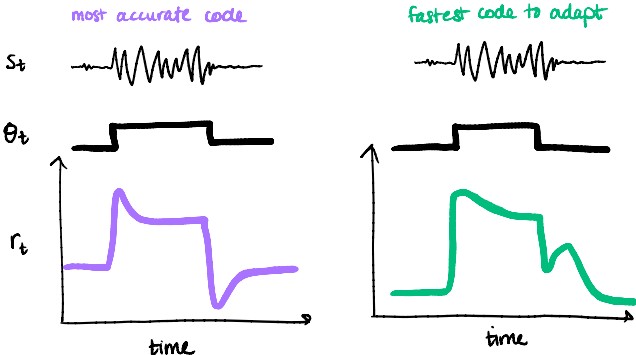

Figure 8: Optimal neural dynamics exhibit qualitative features of "adapting" (left) and "sensitizing" (right) retinal ganglion cells in response to switches in stimulus variance (adapted from [18]).

As highlighted in the previous discussion, both of these strategies are optimized greedily based on the observer's current knowledge of the environment, and exhibit different tradeoffs in performance. One might therefore wonder whether it is possible to leverage the advantages of both strategies. To explore this question, one can form composite codes that balance these two objectives:

$$\operatorname{argmin}_{k,s_0} \alpha \left\langle (\hat{s} - s)^2 \right\rangle_{P(s_t|\hat{\theta}_t)} + (1-\alpha) \left\langle \left( \hat{\theta}(s) - \hat{\theta}(\hat{s}) \right)^2 \right\rangle_{P(s_t|\hat{\theta}_t)}, \tag{11}$$

where $\alpha \in [0,1]$ is a parameter that balances reconstruction and inference errors. For many different assumptions about the stimulus distributions and their dynamics, the optimal encoder balances these two sources of errors, with an optimal value of $\alpha$ that is between 0 and 1 (see [18] for specific examples). This finding suggests that optimally reconstructing signals in nonstationary environments requires devoting some bandwidth to detecting changes in the underlying distribution of those signals.

This also provides a normative perspective on the temporal dynamics of sensory adaptation (Figure 8). Sensory codes that are optimized for reconstruction versus inference show different temporal dynamics that mimic those observed in so-called adapting and sensitizing retinal ganglion cells in the salamander, mouse, and primate [22, 25, 26]. Moreover, this framework predicts that slower environmental dynamics (smaller values of $h$) will lead to slower adaptation, because the ideal observer requires more stimulus samples to be convinced of a change. As a result, the predicted timescale of adaptation scales with the periodicity of changes in the environment, consistent with experimental observations in the fly, mouse, rat, and electric fish [17, 21, 27, 28].

## 2.3 Action selection

*This section covers material from Ma & Hermundstad (2024).*

In the previous sections, we saw how compressed sensory signals (conveyed through a resource-constrained encoder) could impact the inference of latent environmental states, and how this inference, in turn, could be used to dynamically adapt the encoder over time. We formulated this problem in a passive setting, where we did not consider how the inference process impacted the downstream selection of actions. We will now consider an active setting, where inferences guide the selection of actions, which in turn impacts which sensory signals will be gathered in the future.

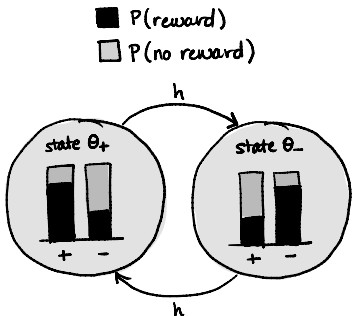

Figure 9: In a example of a nonstationary two-armed bandit task, an agent can sample one of two reward ports ('+' and '−') whose reward probabilities are determined by a latent state $\theta$ that can switch over time.

We will again consider a scenario in which the environment can take on one of two different states, $\theta_t \in \{\theta_+, \theta_-\}$, that can change with a small but fixed probability $h$ per timestep, as given in Eq. 4 (note that we have changed our notation from $H, L$ to $+, -$ for convenience). However, rather than using this state to specify the parameters of a stimulus distribution, we will take this state to specify the probability of rewards at two different ports that can be sampled by an agent (Figure 9). This is an example of a "nonstationary two-armed bandit" task (more generally referred to as a dynamic foraging task) in which an agent is faced with two different levers, or "bandits", that deliver rewards with different probabilities. In a nonstationary setting, these probabilities change over time, and thus the agent is best served by using its past observations to infer which of the two levers is more rewarding at a particular time. In the simplest version of this task, the observations $o \in \{o_+, o_-\}$ are binary: with each lever pull, the agent either receives a reward ($o = o_+$) or receives nothing ($o = o_-$). This type of task has been used to study decision making in many different species, including humans [29, 30], non-human primates [31, 32], rodents [30, 33–36], and flies [37].

At each timestep, the agent has the option of selecting one of two actions, $a \in \{a_+, a_-\}$, that correspond to sampling one of the two levers. When the environment is in state $\theta_+$, the '+' lever is more rewarding, and thus the agent should select action $a_+$; when the environment is in state $\theta_-$, the '−' lever is more rewarding, and the agent should select action $a_-$. We assume that the more rewarding port delivers reward with probability $p_{\text{high}}$, and that the less rewarding port delivers reward with probability $p_{\text{low}}$. The reward probability can then be written as:

$$p(o \mid \theta, a) = \frac{1 + o\,(\theta\, a\Delta p + \Delta\overline{p})}{2}, \tag{12}$$

where $\Delta p = p_{\text{high}} - p_{\text{low}}$, $\overline{p} = (p_{\text{high}} + p_{\text{low}})/2$, and $\Delta\bar{p} = 2\bar{p} - 1$, and where we make use of the following:

$$\begin{aligned}
a &\in \{a_-, a_+\} = \{-1, 1\}, \\
o &\in \{o_-, o_+\} = \{-1, 1\}, \\
\theta &\in \{\theta_-, \theta_+\} = \{-1, 1\}.
\end{aligned} \tag{13}$$

This task is an example of a partially observable Markov decision process, or POMDP. The optimal actions in this task can be derived via two separable steps: (1) using Bayesian inference to derive the ideal observer that optimally infers a belief $u$ about the current environmental state, and (2) using reinforcement learning to derive the optimal behavioral policy $\pi_*(a|u)$, which specifies the optimal actions given the current belief [38–40] (see Appendix D for a brief primer on reinforcement learning; for a more in-depth treatment, see Massimo Vergassola's lectures from the same 2023 Les Houches Summer School in Theoretical Biological Physics).

As in the previous section, we can derive the ideal observer model that optimally infers the current environmental state based on its prior knowledge. We assume that the observer knows (1) that the environment exists in one of two states (+ or −); (2) the values of $p_{\text{high}}$ and $p_{\text{low}}$; and (3) the switching probability $h$. The goal of the inference is then to use the history of past observations $o_{\tau \leq t}$ and actions $a_{\tau \leq t}$ to determine the current state $\theta_t$. Following the derivations in Appendix C, we can write:

$$
\begin{aligned}
p(\theta_t \mid a_{\tau \leq t}, o_{\tau \leq t}) &= \frac{1}{\Omega} p(o_t|\theta_t, a_t, \cancel{a_{\tau < t}, o_{\tau < t}}) p(\theta_t | \cancel{a_t}, a_{\tau < t}, o_{\tau < t}) \\
&= \frac{1}{\Omega} p(o_t|\theta_t, a_t) \sum_{\theta_{t-1}} p(\theta_t|\theta_{t-1}) p(\theta_{t-1}|a_{\tau < t}, o_{\tau < t}) \\
&= \frac{1}{\Omega} \sum_{\theta_{t-1}} \underbrace{p(\theta_t|\theta_{t-1})}_{\text{world dynamics}} \underbrace{p(o_t|\theta_t, a_t)}_{\text{reward delivery}} \underbrace{p(\theta_{t-1}|a_{\tau \leq t-1}, o_{\tau \leq t-1})}_{\text{prior at time } t-1}.
\end{aligned}
\tag{14}
$$

To simplify the final expression of the belief, we define $u_t$ to be the difference in belief values between the two states:

$$
u_t \equiv P(\theta_t = \theta_+ | o_{\tau \leq t}, a_{\tau \leq t}) - P(\theta_t = \theta_- | o_{\tau \leq t}, a_{\tau \leq t}).
\tag{15}
$$

Using this, together with the results from Eq. 12 for $p(o_t|\theta_t, a_t, o_t)$ and Eq. 4 for $p(\theta_t|\theta_{t-1})$, we can write the belief as:

$$
u_t = (1 - 2h) \cdot \frac{a_t \, o_t \, \Delta p + (1 + o_t \Delta \overline{p}) \, u_{t-1}}{a_t \, o_t \, \Delta p \, u_{t-1} + (1 + o_t \Delta \overline{p})}.
\tag{16}
$$

Given the belief $u$, we can then determine the optimal value function $v_*(u)$ that specifies how 'good' it is (in terms of future accumulated rewards) to maintain a particular belief $u$ (see Appendix D for a discussion of value functions). We do this using a technique called value iteration [40] that iteratively updates the value function over time based on the outcomes of different actions (note that we first must discretize $u$ before performing value iteration):

$$
v_t(u) = \max_a \underbrace{\sum_{u', o} p(u', o \mid u, a) \left[ o + (t-1) v_{t-1}(u') \right] / t}_{q(u,a)}.
\tag{17}
$$

Note that this is modified from standard value iteration to include a running average of expected reward [41]; see Eq. D.11 for the standard form. Analogously to the value function $v(u)$, the action-value function $q(u, a)$ quantifies the value of taking a particular action $a$ given a particular belief $u$. These value functions stabilize to their optima relatively quickly; we can then use them to specify the optimal policy:

$$
\pi_*(a|u) = \text{argmax}_a \, q_*(u, a) = \text{sgn} \, u.
\tag{18}
$$

Note that this corresponds to a purely greedy policy given the current belief:

$$
\begin{aligned}
a_{\text{greedy}} &\equiv \underset{a}{\text{argmax}} \langle r(o \mid u, a) \rangle \\
&= \underset{a}{\text{argmax}} \, p(o = o_+ \mid u, a) \\
&= \underset{a}{\text{argmax}} \frac{1 + a \, \Delta p \, u + \Delta \overline{p}}{2} \\
&= \text{sgn} \, u.
\end{aligned}
\tag{19}
$$

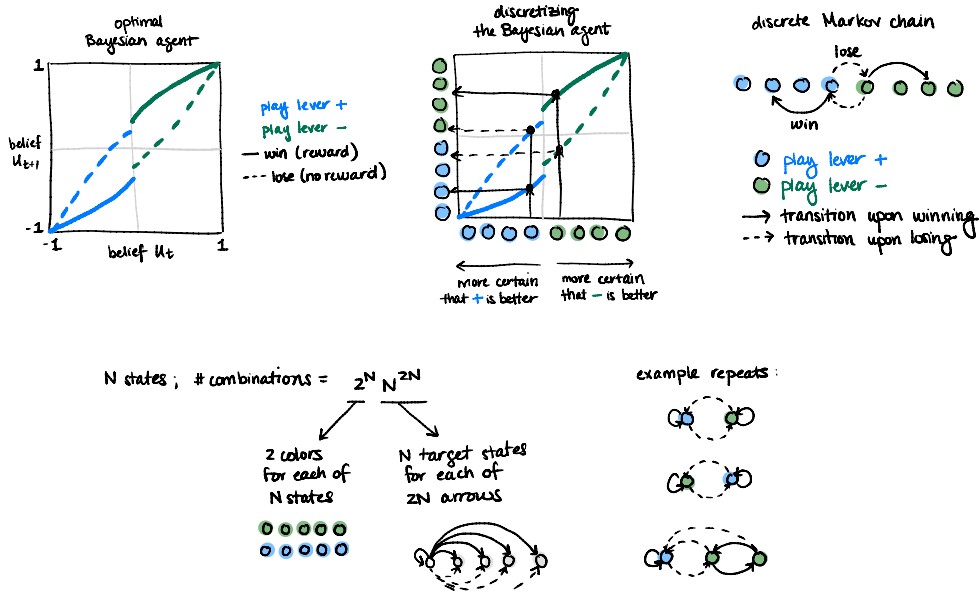

Figure 10: Top row: we can create a compact agent by transforming an optimal Bayesian agent into a discrete Markov chain that specifies actions conditioned on past outcomes. Bottom row: by varying the elements of the Markov chain, there are many ways to build compact agents that achieve good performance (adapted from [41]).

This illustrates how we can use an optimal belief, derived through Bayesian inference, to determine the optimal actions conditioned on that belief. In contrast to the previous section, here we assume that the agent can precisely measure observations in the environment (a reasonable assumption if, as in this case, the observations are binary). However, this also assumes that agent can precisely track its current belief. Just as we explored how the compression of incoming sensory signals impacts inference, so too can we consider how the compression of an internal belief can impact the action selection process. However, as soon as we attempt to compress the belief $u$, there is no guarantee that the optimal (compressed) policy can be derived via the two separable steps of deriving the ideal observer model and deriving the optimal policy conditioned on the observer model. In fact, there are many compressed policies that achieve near optimal performance [41] (see [42] for a general treatment of the tradeoff between optimal performance and computational complexity).

To see this, consider discretizing the belief update in Eq. 16 under the optimal policy in Eq. 18 (Figure 10). This update can be transformed into a Markov chain that consists of $n \in N$ states corresponding to each of the discretized belief values. These states can be labeled according to the actions that they specify; all states with $u(n) > 0$ are labeled $+$, and all states will $u(n) < 0$ are labeled $-$. There are $2N$ transitions between states, corresponding to the two observations that can be obtained from taking the action specified by each state; these transitions can be determined from Eq. 16. In the limit as $N \to \infty$, this Markov chain will approximate the optimal belief update and action selection. However, in the limit that $N$ is finite, the Markov chain can deviate significantly from the optimum, depending on its architecture. For $N$ small, we can directly enumerate all possible Markov chains and compare their performance. For $N$ states, there are at most $2^N N^{2N}$ possible Markov chains (corresponding to 2 labels for each of $N$ states, times $N$ target states for each of $2N$ possible transitions). For $N = 5$ states, this amounts to a whopping $312,500,000$ possibilities. Luckily, $99.9\%$ of these

are repeats that generate identical behavior under the symmetric task that we've considered; a total of $263,428$ are unique. This is still a staggering number; again, luckily for us, only a fraction of these—on the order of $5,000$—exhibit high performance. Of these, only a small fraction exhibit behavior that can be interpreted analogously to the belief update of a Bayesian agent [41]. Thus, by considering compact systems that have limited resources for performing inferences and guiding actions, we can discover a myriad of solutions that achieve good performance.

## 2.4 Summary

In this section, we have introduced some key normative frameworks—including efficient coding (Section 2.1), Bayesian inference (Section 2.2), and reinforcement learning (Section 2.3)—that have been used to understand optimal sensory coding, inference, and action selection. These frameworks allow us to understand the nature of computations that generalize across species, brain regions, and modalities. We have also seen examples of how these frameworks inform and constrain one another: for example, in Section 2.2, we saw how different sensory coding schemes impact the process of inference, and how inference, in turn, can be used to guide adaptive coding schemes; in Section 2.3, we saw how optimal inferences can be used to select actions in uncertain and changing environments, and how compact representations of those inferences can yield a diversity of different strategies for guiding effective behavior.

In the next section, we will use these approaches to understand neural dynamics and behavior in a specific system, and we will follow a set of computations from the encoding of sensory stimuli to the selection of actions (and back).

# 3 Closing the loop in a single system

We now consider a specific system—visually-guided navigation and learning in the fruit fly—where we can, in principle, follow an entire stream of computations, from the encoding of sensory stimuli, to their use in building and modifying internal representations of stored variables, to the selection of actions based on these internal representations. Flies have a reasonably rich behavioral repertoire that allows us to study how behavior changes over time in response to new experience. Moreover, there is a massive genetic toolkit that enables us to monitor and manipulate specific cell types during behavior. Finally, with the recent release of a synaptic-level connectome, it is now possible to relate the physiology of these cell types to their morphology and connectivity.

For the purposes of this discussion, we will consider a class of behaviors in which flies associate rewards and punishments with stimuli that signal different locations in the environment, and use these associations to modify their behavior over time, for example to navigate to good locations or away from bad ones. As an agent in such an environment, this type of navigation is enabled by knowing (A) where you are, (B) where you want to go, and (C) how to get from A to B. In what follows, we will consider how these quantities are represented in the brain and updated based on experience.

## 3.1 Using sensory stimuli to build accurate internal representations

*This section covers material from Noorman, Hulse, Jayaraman, Romani, & Hermundstad (2022) and Kim, Hermundstad, Romani, Abbott, & Jayaraman (2019); see also: Fisher, Marquis, D'Alessandro, & Wilson (2019).*

We begin by asking how an animal knows where it is—i.e., how the brain keeps track of where

an animal is relative to its surroundings. This is something that many animals do effortlessly. For example, if I asked you to close your eyes, and with your left index finger, point to the door that you used to enter the room you're currently in—you would be able to do this without any external sensory signals, and without knowing in advance that this is something you'd have to remember upon entering the room. Our brains use sensory signals to build maps of our surroundings, and we can use these maps 'offline' when we no longer have direct access to those signals.

For several decades now, we have had a theoretical framework that explains how a network of neurons might build and maintain an internal sense of direction [43–51] (see [52–55] for recent reviews). The theory of 'attractor networks' posits that a population of neurons can encode and update internal representations of a variable (such as which direction you're facing) through their recurrent activity. To give some intuition for how this could work, consider building a neural compass, made from a population of neurons, that keeps track of your current direction. There are a few key requirements that we might have for this compass: (1) we want one single compass needle (no more, no fewer), (2) we want the needle to move when we turn, and (3) we want the needle to stop when we stop turning. We can build such a compass with a population of neurons that respond to different orientations; i.e., when you're facing north, one group of neurons responds, and when you're facing east, another group responds. If we arrange these neurons along a ring based on this tuning, then we can figure out how to connect these neurons together to meet all of the requirements listed above. To build a single compass needle, we can connect nearby neurons on the ring with local excitatory connections...but if all neurons locally excite their neighbors, then any activity in one position along the ring will spread to fill the entire ring, and we will lose our compass needle. To prevent this from happening, we can have neurons broadly inhibit distant neurons around the ring; this will keep any activity localized in one region of the ring. With these two ingredients—local excitation and broad inhibition—we can create a single compass needle in the form of a localized bump of neural activity (Figure 11). This motif of local excitation and broad inhibition is observed in many different biological settings. In cortical circuits, for example, a tight balance between excitation and inhibition is thought to be important for shaping neural response properties [56], and such "balanced networks" have been proposed to enhance coding efficiency and capacity [57]. More broadly, the so-called local excitation, global inhibition (LEGI) model has been used to describe how chemotaxing cells can generate an internal (signalling) compass by spatially regulating the activity of signalling pathways [58]. In our context, we use local excitation and broad inhibition to show how networks of neurons can generate an internal (neural) compass by maintaining a persistent bump of neural activity.

To move this bump of activity, we need to invoke additional mechanisms: we can do this by including additional inputs to the ring that are tuned to angular velocity, and that effectively 'push' the bump around the ring when turning right or left. And to keep the bump still when those inputs are removed, we need to ensure that the activity pattern that holds the bump is stable, and can persist at the same orientation along the ring without any external inputs. Theoretically, this typically requires that we use an infinitely large population of neurons to build the compass (see [44, 49, 50, 59] for studies that use large networks to approximate the precision of infinite networks, and [60–62] for studies that highlight failure modes of small networks). However, biology seems to be able to construct a neural compass from only a handful of neurons. The fruit fly, for example, maintains an internal sense of direction in a donut-shaped brain structure called the Ellipsoid Body (EB). A population of "compass neurons" maintains a persistent bump of activity that encodes the fly's current heading [63], and that can be updated by integrating the fly's angular velocity [61, 64] (Figure 12). The connectivity [65, 66] and dynamics [67, 68] of this network are consistent with theoretical accounts of a continuous attractor network, with one notable exception: the fly network is

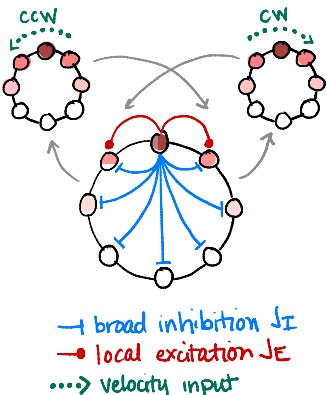

Figure 11: A ring attractor network can maintain a localized bump of neural activity, and can shift the bump by integrating self-motion input.

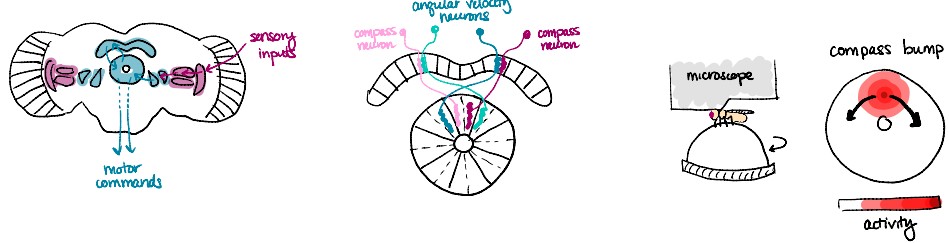

Figure 12: The fly maintains an internal sense of direction in a neural compass. Left: schematic of the fly brain. Middle: schematic of neurons that innervate a donut-shaped brain region called the ellipsoid body (EB). Right: it is possible to image calcium activity of compass neurons in the EB while the fly is tethered under a microscope and walking on an air-lifted ball. The activity of the compass neurons forms a localized bump that moves around the donut and tracks the movements of the fly relative to the world.

composed of a very small number of computational units (on the order of 8 units), rather than the extremely large network that we'd expect to need. In what follows, we will explore how it is possible to accurately build, maintain, and update an internal sense of direction in such a small network.

### 3.1.1 Maintaining persistent internal representations in the absence of input

To see how we go about building such a network, let's begin with a population of neurons, indexed $i \in [1, ..., N]$, that have neural activities $r_1, ..., r_N$. Let's consider the case where we can describe these activity with a linear system of equations:

$$\dot{\mathbf{r}} = -r + W\mathbf{r} + C_0, \tag{20}$$

where $W$ is a connectivity matrix that specifies the strength of excitatory or inhibitory connections between all pairs of neurons, and $C_0$ is a constant that specifies feedforward activity that is injected into the network. To build some intuition, consider the following connectivity matrix:

$$W = \begin{bmatrix} \lambda_1 & 0 \\ 0 & \lambda_2 \end{bmatrix}, \tag{21}$$

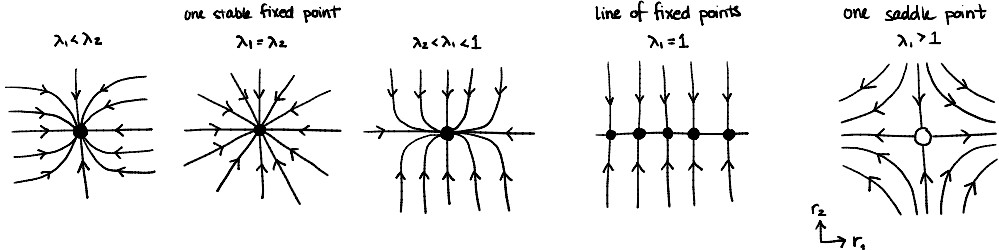

Figure 13: Fixed points of the linear system in Eq. 22, with $\lambda_2 < 0$ (adapted from [69]).

where $\lambda_2 < 0$. This defines a linearly-independent system of equations (adapted from Steven Strogatz's book [69]):

$$\begin{bmatrix} \dot{r}_1 \\ \dot{r}_2 \end{bmatrix} = \begin{bmatrix} \lambda_1 - 1 & 0 \\ 0 & \lambda_2 - 1 \end{bmatrix} \begin{bmatrix} r_1 \\ r_2 \end{bmatrix} + C_0, \tag{22}$$

whose solutions are:

$$\begin{aligned} r_1(t) &\propto A_1 + B_1 \exp\big((\lambda_1 - 1)t\big), \\ r_2(t) &\propto A_2 + B_2 \exp\big((\lambda_2 - 1)t\big), \end{aligned} \tag{23}$$

where $A_1, A_2, B_1, B_2$ are constants. The entries of $W$ determine the dynamics of this system. If $\lambda_1 < 1$, there is one stable fixed point in the network (also called a 'point attractor'); the value of $\lambda_1$ (and specifically whether it is less than or greater than $\lambda_2$), determines the relative rate at which $r_1$ and $r_2$ approach the fixed point (Figure 13). Alternatively, if $\lambda_1 > 1$, there is a single saddle point; the dynamics are stable along the $r_2$ axis, and unstable along the $r_1$ axis. In the special case that $\lambda_1 = 1$, there is a continuum of stable fixed points along the $r_1$ axis (also called a 'line attractor').

In the case where the system of equations is not linearly independent, the same logic applies to the system of equations. The eigenvectors $\xi$ of $\mathbf{W}$ specify the axes of flow, and the sign and magnitude of the eigenvalues $\lambda$ specify the direction and rate of flow, respectively. If the system has a single unity eigenvalue and all other eigenvalues less that one, then there is a continuum of stable fixed points. This continuum can be used to stably encode a single linear variable via appropriate combinations of $r_1$ and $r_2$. This idea was used to propose how a network of as few as two neurons, with activities $r_1$ and $r_2$, could stably encode a linear variable like the position of an animal's eyes [43].

However, this formulation cannot encode a circular variable like orientation. Instead, one can build a *ring* of stable fixed points by stitching together multiple line attractors over fixed intervals (Figure 14). For this to work, there must be a precise handoff between line attractors, which requires a nonlinearity in the network (and some additional fine tuning, which we will come to). We can consider a simple nonlinear network of the form:

$$\dot{\mathbf{r}} = -\mathbf{r} + W\phi(\mathbf{r}) + C_0, \tag{24}$$

where $\phi(\cdot)$ is a nonlinear function. For illustrative purposes, we will take $\phi(\cdot)$ to be a threshold linear function. This form of nonlinearity ensures that only a subset of neurons in the network is active at any given time; importantly, the dynamics of these active neurons are governed by a set of *linear* equations via an active submatrix $\mathbf{W}_{\text{act}}$ of the full connectivity matrix $\mathbf{W}$. As before, the eigenvalues and eigenvectors of this active submatrix determine the fixed point structure of these dynamics. Thus, by appropriately choosing a connectivity matrix $\mathbf{W}$, it is

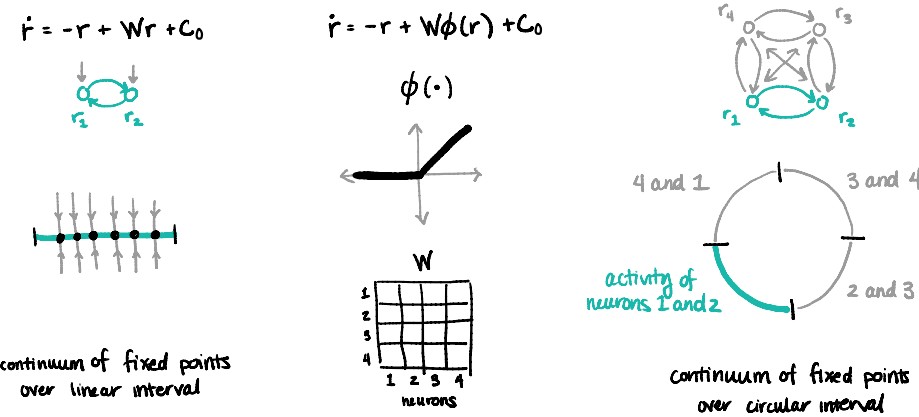

Figure 14: Left: a linear system can encode a continuum of marginally-stable fixed points. Middle, right: in a nonlinear system, a ring attractor can be constructed by stitching together line attractors [68].

possible to ensure that all active submatrices of $\mathbf{W}$ have a single unity eigenvalue, and all other eigenvalues less than one. Each active subsystem can thus be constructed to encode a line attractor over a finite interval, and these line attractors can then be stitched together to form a more complex attractor structure [68].

To construct a ring attractor, we want a rotationally invariant set of line attractors. This can be achieved by choosing $\mathbf{W}$ to be a circulant matrix. A common choice is:

$$\mathbf{W} = \frac{1}{N}\big(J_I + J_E \cos(\theta_i - \theta_j)\big),\tag{25}$$

where $J_I$ controls the broad inhibition in the network, and $J_E$ control local excitation between neurons with preferred headings $\theta_i$ and $\theta_j$. With appropriate choices of $J_E$, the active submatrices of $\mathbf{W}$ are identical, and each has a single unity eigenvalue. For example, given four neurons with activities $r_1$, $r_2$, $r_3$, and $r_4$, a ring attractor can be constructed by stitching together four line attractors spanned by ordered pairs of units: (1) $r_1$ and $r_2$, (2) $r_2$ and $r_3$, (3) $r_3$ and $r_4$, and (4) $r_4$ and $r_1$. In this case, the preferred headings $\theta_i$ and $\theta_{i\pm1}$ are all separated by 90°, and the full connectivity matrix is:

$$W = \frac{J_I}{N} + \frac{J_E}{N}\begin{bmatrix} 1 & 0 & -1 & 0 \\ 0 & 1 & 0 & -1 \\ -1 & 0 & 1 & 0 \\ 0 & -1 & 0 & 1 \end{bmatrix}.\tag{26}$$

The $2 \times 2$ active submatrix has a connectivity structure:

$$W_{\text{act}} = \frac{1}{N}\begin{bmatrix} J_I + J_E & J_I \\ J_I & J_I + J_E \end{bmatrix}.\tag{27}$$

If we diagonalize this active submatrix, the eigenvectors are $\xi_1 = [-1,\, 1]$ and $\xi_2 = [1,\, 1]$, with corresponding eigenvalues $\lambda_1 = J_E/N$ and $\lambda_2 = 2J_I/N + J_E/N$ (note that here, we assume $J_I < 0$, in which case $\lambda_1 \geq \lambda_2$; more generally, it is possible to derive the maximal values of $J_I$ for which the network will generate a stable bump of activity [68]). With $N = 4$ neurons, the leading eigenvalue is 1 if $J_E = 4$. This solution guarantees that the bump of activity can persist anywhere along a continuum of orientations between $\theta_i$ and $\theta_{i\pm1}$ (and one can show that this is a solution for all even-sized networks of size $N \geq 4$; see Figure 15 and [68]).

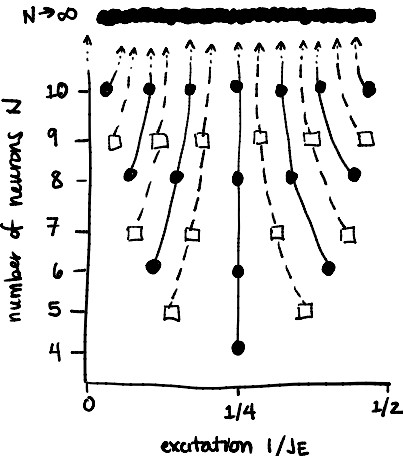

Figure 15: Larger networks have more ways to tune the local excitation in order to generate a ring attractor (adapted from [68]). Circles and squares denote optimal solutions for networks with even and odd numbers of neurons, respectively.

The corresponding eigenvector describes a situation in which any increases in the activity of one neuron are offset by decreases in activity of the other, and the resulting population activity pattern is marginally stable. This type of formulation can be used to construct a ring attractor that can encode a continuum of values on a circle with a localized bump of activity. For a network of $N = 4$, there is one single setting of the local excitation that can generate a continuum of fixed points with line attractors spanning 2 neurons. For a network size $N = 6$, there are 3 ways to generate this continuum from line attractors spanning either 2, 3 or 4 neurons. More generally, for a network of $N$ neurons, there are $N - 3$ ways to generate a ring attractor from line attractors spanning $n = [2, ..., N - 2]$ neurons. Thus, the larger the network, the easier it is to find a parameter setting that will generate a continuum of fixed points around a ring.

This construction assumes that the local excitation, $J_E$, can be chosen optimally. If $J_E$ is not chosen optimally, the full connectivity matrix $W$ will have two different active submatrices that differ in size (Figure 16). The larger of these two active submatrices, with a bump spanned by $N_{act} = n + 1$ active neurons, has a leading eigenvalue greater than one, and thus generates an unstable fixed point; the smaller of these active submatrices, with a bump spanned by $N_{act} = n$ active neurons, has a leading eigenvalue less than one, and generates a single stable fixed point. The bump of activity will then be pushed away from the unstable fixed point and pulled toward the stable fixed point. As it does so, the bump transitions between the two different linear regimes that maintain the bump of activity with $n + 1$ versus $n$ active neurons. The angular span of these two different regimes is closely related to the drift rate within each regime—i.e., how quickly the bump is pushed from or pulled toward a fixed point. As a result, the dynamics of the bump are governed by three factors: the orientations of the stable and unstable fixed points, the rate at which the bump is pulled toward or pushed from these fixed points, and the angular span of each stable and unstable regime. In the limit that $J_E$ approaches an optimal value, the drift rate in one regime tends to zero as the angular span of that regime grows to fill the entire ring. Thus, a ring attractor emerges in the limit that the bump drifts infinitely slowly over an increasingly large fraction of the ring.

Up until now, we have built intuition about this ring attractor solution by patching to-

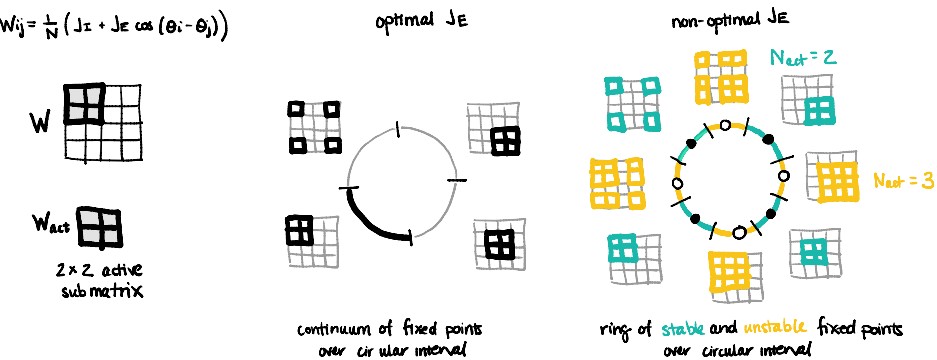

Figure 16: Left, middle: optimal values of local excitation generate a continuum of marginally-stable fixed points on a ring. Right: non-optimal values of local excitation generate a ring of stable and unstable fixed points.

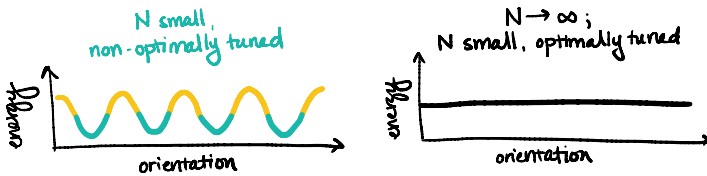

Figure 17: Non-optimal values of local excitation lead to a hilly energy landscape (left); optimal values flatten this landscape (right).

gether linear dynamical systems. An alternative formulation of this solution can be derived by computing the "energy landscape" (a Lyapunov function) of the dynamical system (Figure 17). This energy landscape describes the states to which the system will evolve in the long-time limit. In the limit of an infinitely large network with connectivity given by Eq. 25, the energy landscape will be flat as a function of orientation. For a finite network with naively-chosen $J_E$, the energy will be hilly, with discrete basins (corresponding to stable fixed points) separated by energy barriers (corresponding to unstable fixed points). The optimal values of $J_E$ will flatten this energy landscape. To derive these values, one can first perform a discrete Fourier transform on the system of nonlinear equations in Eqs. 24-25 to transform a system of $N$ equations that describe the dynamics of individual neurons into a system of 3 equations that describe the dynamics of the amplitude $a_c$, width $w_c$, and orientation $\theta_c$ of the compass bump (because of the choice of connectivity, only the DC mode and first modes survive the Fourier transform, after initial transients). This set of equations can then be used to construct the energy of the system. One can then derive the Hessian matrix, which captures the local curvature of the landscape. When computed at the orientations of the stable fixed points, the Hessian separates into a block-diagonal matrix, with a single eigenvector aligned with the orientation $\theta_c$. The corresponding eigenvalue, $\partial^2 E / \partial \theta_c^2$, depends only on $J_E$, $N$, and $N_{\text{act}}$. Thus, for a given network size, it is possible to find the values of $J_E$ that will locally flatten the energy as a function of orientation. One can further show that these same values of $J_E$ agree with those derived from the linear systems perspective, and they guarantee that the energy is *globally* flat as a function of orientation (see the SI of [68] for more details). Our ability to treat this problem analytically relies heavily on the assumption that $\phi(\cdot)$ is a threshold linear function; however, one can numerically find such optimal settings in small networks with other nonlinearities (see SI Figure S3 of [68]).

### 3.1.2 Accurately updating internal representations via internal input

In the previous subsection, we showed how it was possible to maintain a persistent bump of activity at any of a continuum of orientations along a ring. For this bump of activity to function as a compass needle, it must be able to be updated based on changes in orientation. This can be done by injecting self-motion input into the network; the network then has to integrate this input to appropriately shift the bump of activity to the correct location along the ring (note that this self-motion input is thought to be computed from a combination of motor and optic flow inputs [70], which raises an interesting question about whether these inputs are optimally combined to estimate self-motion; see [71] for a perspective on this issue). A simple way to do this is to add two additional "side rings" to the network that are gated by clockwise or counterclockwise angular velocity [72]; these side rings inherit the bump of activity from the primary, or "center" ring, and project back to the center ring with shifted connections. Thus, clockwise turns will push the bump in one direction around the ring, and counterclockwise turns will push the bump in the opposite direction. In the limit that the time constants of these side rings are very fast, we can approximate their dynamics with a set of velocity-dependent inputs into the network:

$$\dot{r}_j = -r_j + \frac{1}{N}\sum_k \left(W_{jk}^{\text{sym}} + v_{\text{in}}W_{jk}^{\text{asym}}\right)\phi\left(r_k\right) + C_0, \quad j = 1,\ldots,N, \tag{28}$$

where $v_{\text{in}}$ is the input velocity, and where

$$\begin{aligned} W_{jk}^{\text{sym}} &= \frac{1}{N}\left(J_I + J_E\cos(\theta_j - \theta_k)\right), \\ W_{jk}^{\text{asym}} &= \frac{1}{N}\sin(\theta_j - \theta_k), \end{aligned} \tag{29}$$

consistent with Eq. 25 (note that we typically include an integration time constant $\tau$, which we have taken to 1). If the excitation is chosen optimally, this network can perfectly integrate its inputs. However, if the excitation is not optimally chosen, the movement of the bump will feel the effects of the stable and unstable fixed points in the network: the bump will move faster than the input velocity as it accelerates away from an unstable fixed point, and the bump will move slower than the input velocity as it decelerates toward a stable fixed point. These dynamics are again determined by the angular orientations of the unstable and stable fixed points, the angular span of the linear regimes about each fixed point, and the drift rate at which the bump is pushed away from or pulled toward these fixed points. In the limit of small velocity inputs, the drift rates and angular spans of each regime will remain unchanged, but the orientations of the fixed points will shift; the stable fixed points will shift in the direction of the input velocity, and the unstable fixed points will shift against the direction of the input velocity. So long as the stable and unstable fixed points remain within their respective regimes, the bump will evolve toward and persist at the stable fixed point, and the network will fail to continuously integrate its inputs. However, above a particular threshold velocity, the stable fixed point will move into the *unstable* regime, and the unstable fixed point will move into the *stable* regime. As a result, when in the stable regime, the bump will be pulled toward a stable fixed point, but can never reach it; instead, it will transition into the unstable regime, where it will be pushed away from an unstable fixed point. This push and pull will cause the bump to speed up and slow down as it transitions between regimes.

In the view of the energy landscape, the drift rates determine the local curvature of the energy landscape within each linear regime. As the velocity increases, the fixed points will move, but the local curvature about those fixed points will remain approximately unchanged. This leads to a tipping of the energy landscape in the direction of the input velocity. If the input velocity is very small, the degree of tipping will be small, and the bump will still get stuck at

the stable fixed points in the network. As the input velocity increases, the tipping will be more severe. Again, above a given threshold velocity, the bump will move continuously down the energy landscape without getting stuck, but will speed up and slow down as it feels the effect of the curvature of the landscape.

### 3.1.3 Reliably tethering internal representations to the external world

The previous subsections highlighted how we could build a network that maintains and updates an internal sense of direction, even in the absence of external sensory cues. However, to function in an external environment, this internal representation must be tethered to cues in that environment. In the fly compass network, these external sensory cues reach the compass through another set of "ring neurons" [66]. There are many different classes of ring neurons that bring in information about polarized light [73–75], wind direction [76], and local visual features [77–80]. This again raises an interesting question as to whether and how these different inputs are combined to (optimally?) infer an estimate of orientation, something we will not discuss in detail here. For the sake of illustration, we will focus one modality that is encoded by visual ring neurons that respond to local spatiotemporal features in a scene [77, 79]. The response properties of these neurons are well captured by Gabor-like spatial filters combined with biphasic temporal filters, a structure that is thought to efficiently exploit the structure of spatiotemporal correlations in natural scenes (as discussed in Section 2.1; [81, 82]). At the simplest level, this means that a given ring neuron will be active whenever a particular feature in the visual scene is present at a particular location relative to the fly. The presence of multiple inputs to the compass—here, external sensory inputs and internal self-motion inputs—raises the question as to how the compass network can reliably keep these sets of inputs in register to accurately update a single, self-consistent internal representation of heading.

This is thought to be achieved through plasticity between ring neurons and compass neurons [83–86]. Each ring neuron makes all-to-all synapses onto compass neurons [66], and these synapses are plastic and can be modified over time based on experience [85–87]. As a result, plasticity is thought to "map" the external world onto the compass, and several lines of ongoing work are focused on how this map remains self-consistent [85–88]. One set of ideas stems from Kohonen's 'self-organizing map', which is an unsupervised competitive learning algorithm that tries to iteratively map a high-dimensional input space into a low-dimensional 'map space' [89] (Figure 18). Given an input space spanned by input patterns $\{p\}$ and a map space spanned by output patterns $\{q\}$, this is achieved through plasticity in weights $W_{ij}$ that link the two:

$$\Delta W_{ij} = \alpha \theta(j)(p_i - W_{ij}). \tag{30}$$

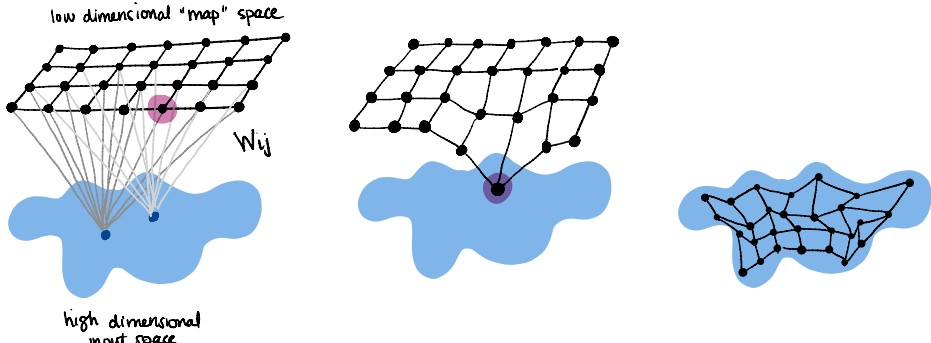

Figure 18: Illustration of self-organizing map developing over time (adapted from Wikipedia, user:Mcld).

For a given input pattern **p**, this rule updates the weights in the local neighborhood $\theta(j)$ around the most strongly responding output unit $q_j$ (also referred to as the 'best matching unit'). The parameter $\alpha$ is a learning rate that determines how quickly the weights will update over time.

If we define the input patterns to be the responses of the ring neurons, and the output patterns to be the responses of the compass neurons, then the ring attractor enforces a natural neighborhood function defined by the bump of compass activity. The rule then becomes:

$$\Delta W_{ij} = \alpha r_j^c (r_i^r - W_{ij}), \tag{31}$$

where $r_i^r$ and $r_j^c$ define the activity of the ring and compass neurons, respectively. This rule is similar to a classic Hebbian learning rule in which neurons that "fire together, wire together"—in other words, coactivity between neurons $i$ and $j$ will lead to a strengthening of the weights that link them.

In the case of the fly's neural compass, the ring neurons are thought to be inhibitory [90, 91], and so the appropriate learning rule is "anti-Hebbian" (i.e., coactivity between neurons $i$ and $j$ will weaken, rather than strengthen, the weights that link them). Moreover, evidence suggests that the learning rate $\alpha$ scales with the fly's angular velocity $v$ [85, 87], and thus has a form similar to:

$$\Delta W_{ij} = -v^2 r_j^c (r_i^r - W_{ij}). \tag{32}$$

This velocity-dependent learning rate ensures that the map is only updated when the fly turns, and does not update when the fly persists at the same orientation for longs periods of time.

As the fly explores a new visual scene, this plasticity rule will generate a self-consistent mapping of the visual world onto the compass, such that turns that drive the bump of activity are matched by weak inhibition from ring neurons that respond to the corresponding features in the visual scene (Figure 19). For example, consider a visual scene that consists of a single visual feature, like a vertical bar. And consider that a given ring neuron $r_i^r$, which responds to features at an orientation $\phi_i$, is co-active with a given compass neuron $r_j^c$, which responds

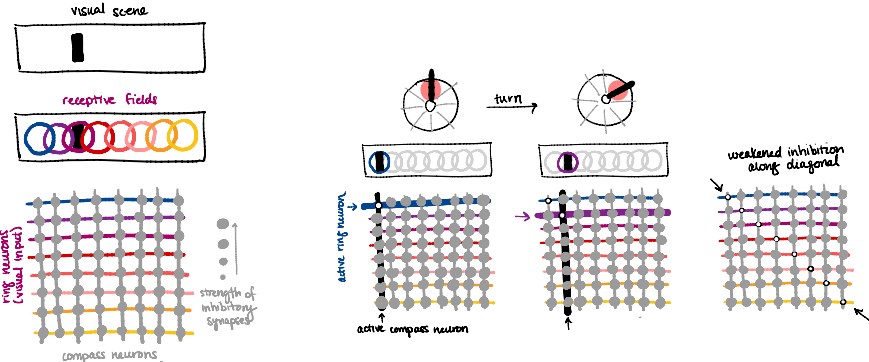

Figure 19: Plasticity creates a self-consistent mapping between visual and self-motion inputs that drive the compass (adapted from [85]; see also [86]). Left: visual ring neurons have receptive fields that tile visual space. These neurons make all-to-all inhibitory synapses onto compass neurons that maintain the compass bump of activity. Middle, right: as the fly turns, the compass bump moves around the EB, exciting a different compass neuron (upper row). At the same time, the visual scene moves relative to the fly, and excites a different ring neuron (middle row). The coactivity between ring and compass neurons weakens the synapses between them. Over time, this creates a self-consistent mapping between the visual and self-motion inputs that drive the compass.

when the bump is at an orientation $\theta_j$. If the fly turns, the visual scene will change by an orientation $\Delta\phi$, such that a ring neuron $i + \Delta\phi$ is now active. At the same time, self motion inputs will drive the bump to change orientations by some angle $\Delta\theta$, such that a compass neuron $j + \Delta\theta$ is now active. The plasticity rule above will ensure that ring neuron $i + \Delta\phi$ will have the weakest weight, and thus the smallest inhibition, onto compass neuron $j + \Delta\theta$. As a result, the compass bump will move in lock step with the movements of the fly and the movements of the visual scene.

By this same logic, one can easily construct a scenario that might alter this mapping. For example, consider a visual scene with a two-fold symmetry, such as two identical bars separated by 180°. This scene will induce identical responses in two ring neurons that respond to orientations separated by 180°. However, the plasticity rule introduced above will eventually weaken synapses at only one location in the compass network, leading the bump to favor one of the two orientations that are associated with this symmetry. I will leave it to the reader to speculate as to how this might impact the dynamics of the compass bump; we will return to this idea in Section 3.3.

## 3.2 Using internal representations to guide effective behavior

*This section covers material from Dan, Hulse, Kappagantula, Jayaraman, & Hermundstad (2024).*

We can now consider how this internal sense of direction is used to guide the fly's behavior. For this, we will start by analyzing the behavior itself, and those changes in behavior that arise during learning. We will then explore how these changes in behavior could be mediated by internal representations in the brain, such as the sense of direction that we have been discussing.

For this, we will consider a variant of a older learning paradigm [92] in which flies navigate in a one-dimensional virtual environment (Figure 20; physicist readers might find it interesting to know that these experiments were pioneered by Martin Heisenberg, the son of Werner Heisenberg, prompting some to ask whether one can formulate an analogous 'fliesenberg uncertainty principle' [93]). This environment consists of an LED screen than spans 330° and displays different visual patterns [94, 95]; this screen surrounds the fly, which is tethered in place but able to move its wings. The orientation of the patterns on the screen is then coupled to the fly's movements; when the fly tries to turn left, the patterns on the screen are rotated to the right, and vice versa. In this way, the fly has closed-loop control of the movements of the patterns on the screen. By tethering the fly, it is then possible to image from the brain while the fly is navigating. By further pairing orientations of the visual scene with rewards and punishments, it is possible to study how the fly's neural activity and behavior change in response to different types of feedback.

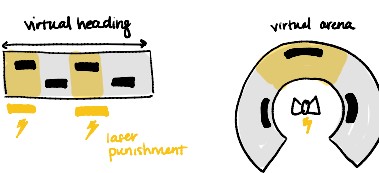

Figure 20: Flies can learn to avoid punishment associated with different visual patterns (adapted from [96]). Left: a visual scene has repeating sets of visual patterns; one set of patterns is paired with an aversive heat punishment. Right: in a virtual reality setup, the fly orients relative to the visual scene, and is pinged in the abdomen with an infrared laser whenever it orients toward one of the punished visual patterns.

### 3.2.1 Inferring a generative model of behavior

Flies exhibit different modes of patterned behavior. In flight, they exhibit periods of straight flight, or 'fixations', punctuated by abrupt turns, or body 'saccades' [94, 96–98] (Figure 21). We can segment the behavior into these two different modes, measure how the properties of these modes vary over time and across flies, and use this to infer a generative model of behavior. This generative model can be phrased in terms of a behavioral policy $\pi(a|\theta)$ that specifies the probability of taking an action $a$ given a particular state $\theta$ of the environment [40] (analogously to our discussion of behavioral policies in Section 2.3).

We begin by assuming that behavior can be structured as a sequence of saccades and fixations, such that the termination of one mode initiates the other mode, and vice versa. In the language of reinforcement learning, each action then comes in two different types: $a \in \{F, S\}$. Each mode can be specified by an angular velocity, $\omega$, and a duration conditioned on that angular velocity, $\Delta t|\omega$.

Fixations tend to have near-zero angular velocity, so we can approximate this as a delta function: $P(\omega) = \delta(\omega)$. The distribution of fixation durations—whether measured across time or flies—is well fit by an inverse Gaussian distribution:

$$P(\Delta t; \mu, \lambda) = \sqrt{\frac{\lambda}{2\pi\Delta t^3}} \exp\left(-\frac{\lambda(\Delta t - \mu)^2}{2\mu^2 \Delta x}\right). \tag{33}$$

This distribution is appealing because we can specify a process that generates it. Consider the following stochastic process:

$$X_t \sim \nu t + \sigma W_t, \tag{34}$$

where $W_t$ is standard Brownian motion, and $\nu$ is a drift term (this process is often called a 'drift diffusion' process). The first passage time of this process, given a fixed threshold $\rho > 0$, follows an inverse Gaussian distribution, with $\mu = \rho/\nu$ and $\lambda = (\rho/\sigma)^2$ [99–101]. Thus, we can capture a single fixational event by sampling from this process; the fixation terminates when the process crosses a threshold $\rho$.

This process requires specifying three parameters: the threshold $\rho$, the drift rate $\nu$, and the diffusion spread $\sigma$. In principle, any or all of these parameters could vary over time based

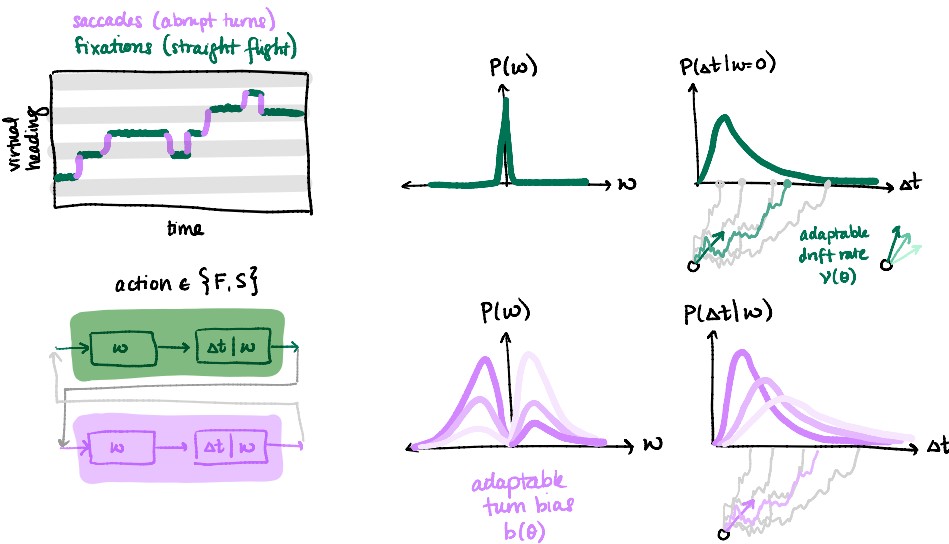

Figure 21: Left: flies fixate (F) and saccade (S). These flight patterns vary in angular velocity $\omega$ and duration $\Delta t$. Right: we can use the statistics of these flight patterns to infer a generative model of behavior (adapted from [96]).

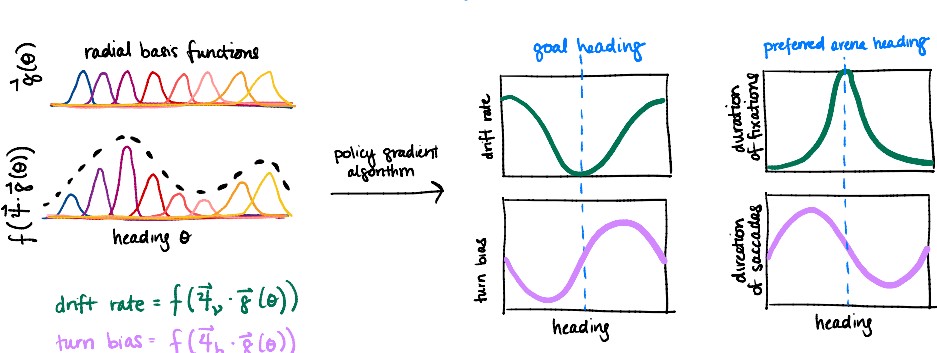

Figure 22: We can use reinforcement learning methods to learn an optimal behavioral policy, and compare it to the behavior of real flies (adapted from [96]).

on experience, and the variation in each of these parameters would manifest in predictable changes in the mean and variance of the inverse Gaussian distribution. To see this, note that the mean mean $= \mu$ and variance var $= \mu^3/\lambda$ are related by: $\log(\text{var}) = 3\log(\text{mean}) - \lambda$. If, for example, the variability in behavior is explained by changes in the drift rate $\nu$ alone (with fixed threshold $\rho$ and spread $\sigma$), we would expect to see the relationship between the variance and mean of fixation durations to follow a line with a slope of 3 and offset of $-\lambda = -\rho^2/\sigma^2$. By fitting this distribution across time and across flies (or, alternatively, by measuring the mean and variance empirically), we can show that the behavior is indeed most consistent with a model in which the drift rate is changing over time. Thus, we can model fixations as events with near-zero angular velocity, and with durations generated by a drift diffusion process with an adaptable drift rate.

Saccades follow a similar structure, but on a different time scale. In contrast to fixations, these events are thought to be ballistic, such that they cannot be interrupted or modified during their execution. Using the same types of analyses, we can show that the angular velocity of saccades is well fit with two log-normal distributions (corresponding to clockwise and counter-clockwise saccades), but with an adaptable bias that captures the tendency to initiate saccades in one direction versus another. We find that the duration of saccades, analogously to fixations, also follows an inverse Gaussian distribution. However, rather than having a flexible drift rate, the distribution of saccades is consistent with an angular-velocity-dependent drift rate. Thus, we can model saccades as events with nonzero angular velocity drawn from a lognormal distribution with an adaptable bias, and with a duration generated by a drift diffusion process with an angular-velocity-dependent drift rate. The similarity in the statistical structure between fixations and saccades suggests that, in both cases, a drift diffusion process can be used to keep time and thereby specify the duration of different behavioral modes. The parameters of that process can then be used to adjust these durations based on other kinematic properties, or based on experience.

### 3.2.2 Learning the parameters of a generative model based on experience

Given a parameterized generative model of behavior, or behavioral policy $\pi(a; \vec{\psi})$, we can now ask how the parameters $\vec{\psi}$ of that policy *should* be modified over time based on experience. To this end, we can construct an agent that comes embodied with this generative model, and the study how the behavior of that agent changes as we update the generative model based on experience (Figure 22).

In problems of spatial navigation, a common approach is to discretize the environment into states $\theta$, and construct a lookup table $q(\vec{\psi}, \theta)$ that specifies the optimal parameter settings for each state. In our setting, these states correspond to different orientations of the agent. Moreover, since we are modeling an agent that uses a neural compass to maintain an internal representation of its own orientation, we will take these states to correspond to the compass heading $\theta_c$ (see Section 3.1 for a discussion of the compass heading). The agent then uses the current parameter settings, together with its own compass heading, to select actions, and then iteratively updates the parameters based on the outcomes of those actions in the current environment. This process can be quite slow, because there is typically no *a priori* structure built into the lookup table, and thus the agent has to experience the consequences of each parameter setting at each compass heading in order to make adjustments to these parameters (see Appendix D for more details).

To speed up this process, the agent can instead learn a continuous function that specifies the parameter settings over space. For example, rather than specifying parameter settings at each of a set of discretized compass headings, one can use a discrete set of continuous basis functions to tile the space of headings, and learn a discrete set of weights on those basis functions (Figure 22). This can speed up the learning process because any parameter changes that are learned at one heading are immediately used to update parameters at nearby headings (if the basis functions are localized in space).

We can use this to learn a policy with the following structure:

$$v(\theta_c; \vec{\psi}_v) = f_v\big(\vec{\psi}_v \cdot \vec{g}(\theta_c)\big), \tag{35}$$

$$b(\theta_c; \vec{\psi}_b) = f_b\big(\vec{\psi}_b \cdot \vec{g}(\theta_c)\big), \tag{36}$$

where $v$ and $b$ specify the drift rate of fixations and turn bias of saccades derived in the previous section, and $\vec{g}(\theta_c)$ specifies a set of basis functions that tile compass headings $\theta_c$. Informed by what we know about the fly's internal compass, we choose a set of 8 von Mises functions; the shape of this function closely matches the shape of the compass bump, and the number mimics the known discretization of the compass (see Section 3.1). We can then use a common policy-gradient algorithm to update the policy parameters based on experience (see Appendix E for a derivation of this algorithm):

$$
\begin{aligned}
&\text{sample action from policy}\\
&\quad a \sim \pi(\cdot|\theta_c; \vec{\psi}),\\
&\text{take action, gather reward}\\
&\quad R = r(a, \theta_c),\\
&\text{update parameters}\\
&\quad \Delta\vec{\psi} = \alpha R \nabla_{\vec{\psi}} \log\big(\pi(a|\theta_c; \vec{\psi})\big),\\
&\text{update state } \theta_c,
\end{aligned}
\tag{37}
$$

where $r(a, \theta_c)$ is a reward function that we will discuss shortly. Using this algorithm, this agent can be trained through experience to reliably gather rewards or avoid punishments by iteratively updating the parameters of its behavioral policy. We can then compare the output of this trained agent to the behavior of real flies after they have gone through the learning paradigm schematized in Figure 20.

In this paradigm, different orientations of the visual scene are paired with rewards or punishments. The visual scene itself consists of a set of 4 visual patterns, each separated by 90°. The scene was chosen to have a two-fold symmetry, such there are two distinct visual patterns, *A* and *B*, each repeated twice (i.e., *ABAB*). One set of patterns (either *A* or *B*) is paired with an aversive heat punishment that is delivered by pinging the fly in the abdomen

with an infrared laser. Thus, if pattern *A* is punished, the fly would experience heat whenever it orients within ±45° of the center of pattern *A* (we refer to this region as the "danger zone", and the unpunished region as the "safe zone"). Because we do not know whether learning is driven by the negative effects of punishment or the positive effects of relief from punishment, we treat the former as generating a negative reward ($R = -1$), and the latter as generating a positive reward ($R = +1$; note that one can use techniques from inverse reinforcement learning to try to deduce this reward function from observed behavior [102]).

When we train an artificial agent in this same paradigm, the agent learns to generate fixations and saccades who drift rates $v(\theta_c; \vec{\psi}_v)$ and turn biases $b(\theta_c; \vec{\psi}_b)$ vary approximately sinusoidally with the compass heading $\theta_c$ (Figure 22):

$$
\begin{aligned}
v(\theta_c; \vec{\psi}_v) &\propto (1 - \cos(\theta_c - \theta_{\text{safe}}))/2 \,, \\
b(\theta_c; \vec{\psi}_b) &\propto (1 - \sin(\theta_c - \theta_{\text{safe}}))/2 \,,
\end{aligned}
\tag{38}
$$

where $\theta_{\text{safe}}$ denotes the center of the safe zone. Low drift rates lead to long fixations at the center of the safe zone; a high turn bias to the left of the safe zone generates directed turns toward safety.

When we compare this learned behavior to the behavior of real flies, we see that flies exhibit similar structure to their behavior (to see this, we must align the fixations and saccades of individual flies to heading at which these actions were initiated relative to the center of the safe zone, and then average across flies; signatures of this structure can also been seen on an individual fly basis). However, this behavioral structure is apparent in flies even before they experience any laser punishment, provided that we align the behavior to the preferred heading of each fly prior to averaging. The fact that both naive and trained flies exhibit the same behavioral structure suggests that this is not something that needs to be learned in a new environment; rather, this behavioral structure seems to innately guide how flies sample their surroundings. If this is the case, learning need only shift and scale this behavioral structure, rather than build it up from scratch.

Such a formulation—in which the behavioral policy has a fixed functional form, but flexible parameters—dramatically changes the learning process. Rather than building an agent that learns individual associations between policy parameters and locations, this suggests that the agent should come pre-equipped with a policy that specifies functional relationships between policy parameters at all locations in the environment. By building this sort of function into the agent, any changes that are learned at one location can be used to update behavior at all locations (rather than being restricted to the immediate vicinity of the agent). This can significantly speed up the learning process, but comes at the cost of restricting the space of behavioral policies that the agent can learn. Moreover, for this to work in real brains, neural circuits must be able to enforce this behavioral structure while flexibly shifting and scaling this structure to direct movements to different parts of the environment. In the next section, we discuss how the brain might achieve this.

### 3.2.3   Building a structured behavioral policy

To construct the policy given in Eq. 38, we use simple operations that could feasibly be performed by neural circuits (Figure 23). We assume that the fly maintains encodes its current compass heading in a sinusoidal activity profile $f_c(\theta; \theta_c) = \cos(\theta - \theta_c)$, where $\theta_c$ denotes the current compass heading, and $\theta$ describes an anatomical axis that encodes orientation. Note that this profile takes on a von Mises shape in the brain region that first encodes the compass heading, but is reformatted into a sinusoidal shape before propagating to downstream brain regions [65]. This sinusoidal shape is thought to support a range of different vector computations [103].

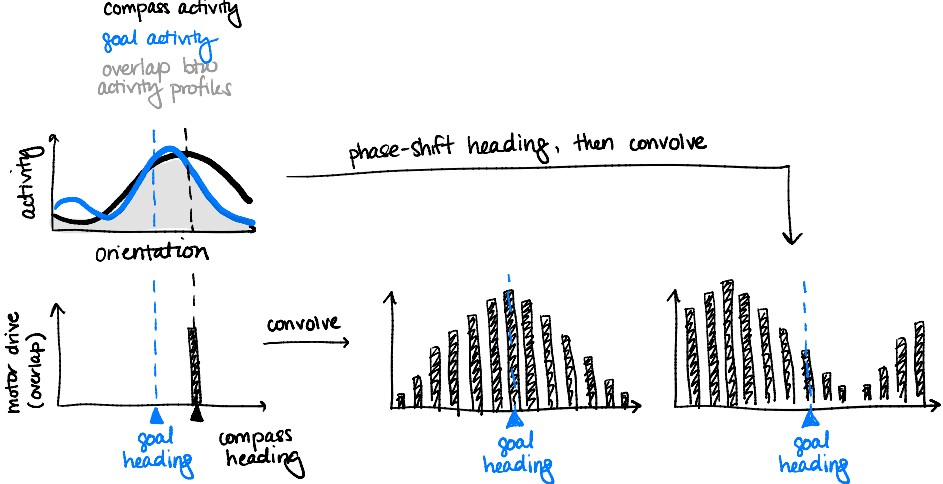

Figure 23: Convolving a sinusoidal compass heading with an arbitrary goal profile leads to a structured motor output (adapted from [96]; see also [104]).

We assume that the fly maintains a goal activity profile $f_g(\theta)$ that can take on an arbitrary shape. When we convolve these two activity profiles—$f_c(\theta;\theta_c)$ and $f_g(\theta)$—the output is sinusoidal and peaks when the compass heading is aligned with the circular mean of the goal activity profile (Figure 23; see Appendix F for brief derivation). We can thus define the goal heading, $\theta_g$, to be the circular mean of the goal activity profile $f_g(\theta)$.

If we use the output of this convolution as a motor drive, this operation will guarantee that the motor drive is strongest when the compass heading is aligned with the goal heading. However, the profiles given in Eq. 38 require motor drives that are peaked 90° and 180° away from the goal heading. This can be achieved by phase-shifting the compass heading (and thus the compass activity profile) before convolving it with the goal activity profile. For a phase shift of $\phi$, the resulting motor drive will be peaked when the compass heading is aligned with $\theta_g - \phi$ (Appendix F). To create the profiles in Eq. 38, we need phase shifts:

$$
\begin{aligned}
\phi_v &= 180°, \\
\phi_b &= 90°.
\end{aligned}
\tag{39}
$$

This formulation guarantees that the motor drive will be sinusoidally structured, regardless of the shape of the goal activity profile, and leads to a policy of the form:

$$
\begin{aligned}
v(\theta_c; \vec{\psi}_v) &\propto A_g \cos(\theta_c - \theta_g + \phi_v) + B_v, \\
b(\theta_c; \vec{\psi}_b) &\propto A_g \cos(\theta_c - \theta_g + \phi_b) + B_b,
\end{aligned}
\tag{40}
$$

where $A_g$ and $\theta_g$ are the strength and orientation of the circular mean of $f_g(\theta)$, respectively, and $(B_v, B_b)$ are offsets. This formulation means that the goal activity profile can be updated over time based on experience, without disrupting the motor drive profile. We can thus use simple Hebbian plasticity rules (analogous to those presented in Section 3.1.3) to update the goal profile over time:

$$
\Delta f_g(\theta, \theta_c; f_g) = \alpha_g \Delta_g,
\tag{41}
$$

where

$$
\Delta_g = \begin{cases}
+\left[f_c(\theta;\theta_c) - f_g(\theta)\right]_+ \Theta(1 - f_g) - \left[f_g(\theta) - f_c(\theta;\theta_c)\right]_+ \Theta(f_g), & R(\theta) > 0, \\
-\left[f_c(\theta;\theta_c) - f_g(\theta)\right]_+ \Theta(f_g) + \left[f_g(\theta) - f_c(\theta;\theta_c)\right]_+ \Theta(1 - f_g), & R(\theta) < 0, \\
0, & R(\theta_A) = 0.
\end{cases}
\tag{42}
$$

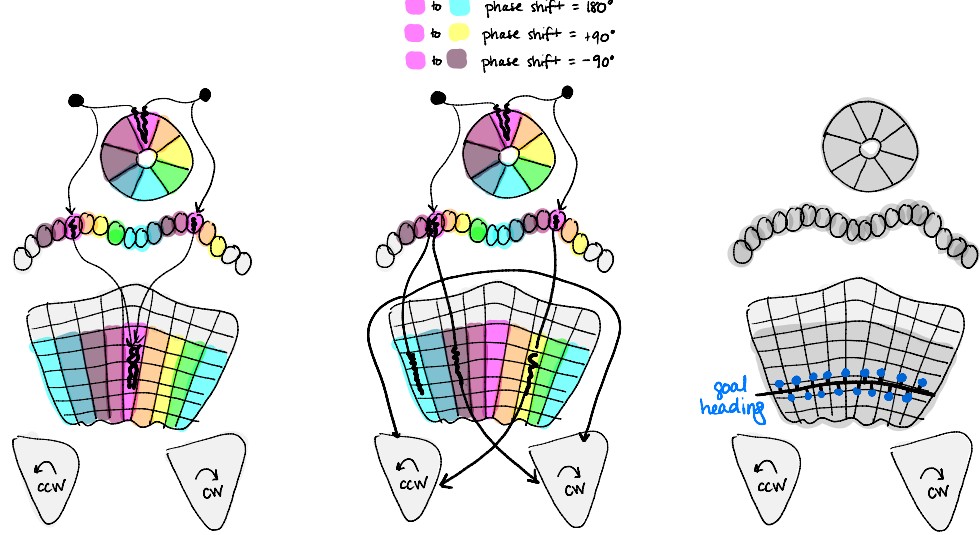

Figure 24: Left: Neurons in the fly central brain implement an anatomical reference frame tethered to the fly's current heading. Center: additional populations of neurons implement anatomical phase shifts in this reference frame. Right: putative goal neurons could store and update goal headings (adapted from [66] and [96]).

Here, $[\cdot]_+$ denotes rectification, and $\Theta(\cdot)$ is the heaviside function. This plasticity rule will strengthen the goal profile at the current compass heading when the fly is being rewarded ($R > 0$), and will weaken the goal profile at the current compass heading when the fly is being punished ($R < 0$).

The fly central brain has circuit motifs that could plausibly implement this set of computations [66, 96] (Figure 24). As discussed in Section 3.1, the compass heading is maintained by a ring attractor network in a region of the brain called the ellipsoid body (EB). From the EB, the compass activity profile is reformatted from a von-Mises-like shape to a sinusoidal shape [65] before traveling to a downstream region called the fan-shaped body (FB) that is thought to carry and combine internal representations about the fly's goals and future actions [66]. There are two populations of neurons—so-called PFL2 and PFL3 populations—that are thought to implement the phase shifts in Eq. 39, respectively, and that project either bilaterally or unilaterally to pre-motor regions that are involved in initiating rightward and leftward turns. The PFL2 population implements a 180° phase shift and projects bilaterally to both premotor regions, and is thus well-positioned to control straight flight. The two PFL3 populations implement ±90° phase shifts and project unilaterally to one or the other premotor regions, and are thus well-positioned to control directed turns. Finally, there are dopaminergic neurons that synapse onto both PFL2 and PFL3 populations, and that themselves receive inputs that are shaped by the fly's current compass heading; thus, these neurons are well-positioned to update the strength of goal synapses in a heading-dependent manner, which could in turn be used to drive behavior through the PFL populations.

## 3.3 Coupling sensory representations and behavior in closed loop

In Section 3.1, we saw how small neural circuits could maintain and accurately update an internal representation of heading, and could tether this representation to sensory cues in the environment through a form of unsupervised learning. In Section 3.2, we saw how this internal representation of heading could be used to update an internal representation of goals

in the environment through a form of reinforcement learning. These two systems—learning where you are in your environment, and learning goals in that environment—are coupled through the same internal representation of heading, and through the actions that are selected by the behavioral policy that is tethered to this representation (Figure 25). As a result, any inaccuracies in this representation will impact the formation of goals, the selection of actions, and the subsequent sensory stimuli that are used to update this representation. As a result, the coupling of these systems can shape individual variability in behavior and learning.

This becomes apparent if we revisit the ideas from Section 3.1.3, about how the internal representation of heading becomes tethered to the outside world. At the end of that section, we introduced the idea that symmetries in the visual scene can lead to a scenario in which plasticity tethers the bump to one of two symmetric location in the environment. If two ring neurons are active for a particular view of a symmetric scene, but one strongly inhibits the bump and the other only weakly inhibits the bump, this induces competition between these two ring neurons, and the bump will tend to jump to the location of weakest inhibition. In other words, the compass will develop a two-to-one mapping, in which two different symmetric orientations of the scene will be mapped onto a single compass heading. This type of confusion is analogous to the confusion you might expect to experience inside a room with two identical doors on opposite sides of the room. When you initially enter the room, you might remember which of the two doors you used to enter, but after some time, you might get confused about which door is which. We can see this confusion in the dynamics of the compass bump, and we can watch the compass bump jump between different orientations that correspond to symmetric views of the visual world [96].

This highlights the important difference between the absolute properties of the visual environment and the relative nature of the internal constructs that we build to represent that environment. In many cases, these are closely matched, but in some cases, our internal perceptions differ markedly from the outside world. This has interesting implications for any downstream behavior that is tethered to these representations. For example, if the fly's actions are tethered to this internal representation, as evidence suggests, then its actions will follow this representation, even as that representation is jumping over time. This can be advantageous in an environment where rewards and punishments are coupled to these same symmetries of the environment. For example, given a policy that is tethered to the difference between the fly's current and goal headings (Eq. 40), a jumping heading bump will serve to

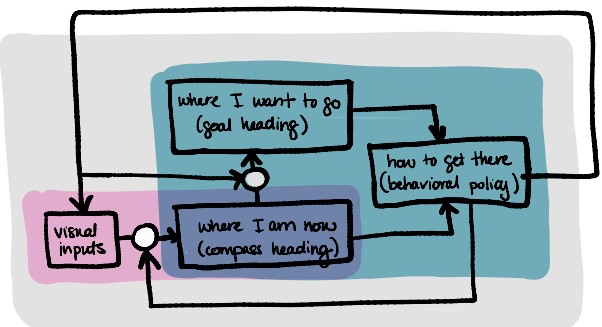

Figure 25: Two learning systems interact to guide behavior. One learning system (pink) maps visual inputs onto an internal compass heading; a second learning system (green) uses the compass heading to learn a goal heading and guide actions via a behavioral policy. Plasticity acts at specific locations (white circles) to allow these representations to change over time based on experience.

copy over this policy at the arena headings that correspond to symmetric views of the visual scene. The means that the fly need only learn one goal heading; its neural circuits will efficiently structure behavior about that goal heading, and its jumping compass bump will copy that structure at multiple locations that share the same visual patterns and reward structures.

For this to work effectively, the mapping of the visual world onto the compass must be stabilized, which takes time in a new environment. As a result, stronger initial goal headings, which lead to more exploitative behavior about the goal heading, can help to quickly anchor the mapping of the visual world onto the compass, particular in the presence of visual symmetries. This, in turn, can help the fly more quickly update its goal heading in the face of rewards and punishments. Weak goal headings, on the other hand, can lead to more diffusive, exploratory behavior, can slow the development of the visual map in the presence of symmetries, and can lead to misalignment between the learning of the visual map and the learning of goals tethered to that visual map. This, in turn, can slow the overall learning process. As a result, individual variability in learning can arise purely because of the dynamics that couple these multiple internal representations. See [96] for further discussion.

## 3.4 Summary

In this section, we explored some of the challenges of building compact internal representations of the outside world, bootstrapping these representations by using one evolving representation to update another, and ultimately tethering efficient behavior to these internal representations. One of the major insights from this line of work is that pre-motor circuits in the brain seems to employ hardwired anatomical motifs for exploiting structure in an animal's predictable relationship to its surroundings, just as early sensory circuits exploit predictable structure in incoming sensory signals. In the field of machine learning, these architectural constraints are often called 'structural priors' or 'inductive biases', and ongoing work in the field seeks to understand what sorts of inductive biases can lead to more flexible and generalizable architectures [105, 106]. Here, we saw how inductive biases in the fly brain—in the form of anatomical motifs that efficiently structure behavior about a single goal heading—can significantly speed learning by reducing the number of associations that an animal has to experience. This suggests an exciting direction for future research, in terms of understanding how the brain should best balance the speed enabled by hardwired circuits, and the flexibility afforded by plasticity within these circuits. More broadly, this prompts the question as to what additional features of an animal's relationship to its environment are predictable enough to warrant hard-wired solutions—something that will be aided by quantitative analyses of natural behavior in the context of ethologically-relevant tasks.

# 4 Outlook

In these notes, we have touched upon different theoretical frameworks—including efficient coding, Bayesian inference, and reinforcement learning—that are used to understand sensory coding, inference, and action selection. We then used aspects of these frameworks in the context of a specific example—navigational learning in the fruit fly—to understand how real brains make sense of incoming sensory stimuli, build internal representations of their relationship to the external world, and use these internal representations to guide behavior. In both of these contexts, we discussed how these frameworks provide a perspective on not only *how* the brain operates, but also *why* it might operate in one manner versus another. Looking ahead, one major opportunity in theoretical neuroscience is to understand how these different frameworks relate to one another, and how they inform and constrain one another. We saw how resource constraints, as considered in the framework of efficient coding, can impact the speed

and accuracy of Bayesian inference, and how constraints on the precision of inference can in turn impact action selection. We also saw how behavioral policies can exploit the reliable statistics of movement to speed learning, just as sensory codes can exploit the reliable statistics of sensory stimuli to speed inference. Finally, we saw that it is becoming possible to test the predictions of these interconnected frameworks in real brains, where the fruit fly—with its targeted genetic tools, near-complete connectome, and behavioral repertoire—provides one exciting test bed to link multiple successive computations in closed-loop.

## Acknowledgements

Wiktor Młynarski and Marcella Noorman provided helpful feedback on these notes. As indicated in the respective section headings, some of the work and perspectives presented in Section 2 derived from collaborations with Wiktor Młynarski and Tzuhsuan (Maz) Ma; similarly, some of the work and perspectives presented in Section 3 derived from collaborations with Marcella Noorman, Brad Hulse, Vivek Jayaraman, Sung Soo Kim, Larry Abbott, Sandro Romani, and Chuntao Dan. Some perspectives presented in the introduction derived from collaborations with Gašper Tkačik, Jonathan Victor, Mary Conte, John Briguglio, and Vijay Balasubramanian.

**Funding information** This work was supported by the Howard Hughes Medical Institute.

## A   Decomposing mutual information

The mutual information between $X$ and $Y$ is defined as:

$$I(X;Y) = \sum_{x \in X, y \in Y} P(x,y) \log\left(\frac{P(x,y)}{P(x)P(y)}\right). \tag{A.1}$$

We can write $P(x,y) = P(x|y)P(y)$, which allows us to re-express mutual information:

$$\begin{aligned}
I(X;Y) &= \sum_{x \in X, y \in Y} P(x|y)P(y) \log\left(\frac{P(x|y)P(y)}{P(x)P(y)}\right) \\
&= \sum_{y \in Y} P(y) \sum_{x \in X} P(x|y)\left[\log P(x|y) - \log P(x)\right].
\end{aligned} \tag{A.2}$$

Noting that the conditional entropy of $X$ given $Y$ is defined as:

$$H(X|Y) = \sum_{y \in Y} P(y)\left[-\sum_{x \in X} P(x|y) \log P(x|y)\right], \tag{A.3}$$

we can use this to rewrite the mutual information as:

$$\begin{aligned}
I(X;Y) &= -H(X|Y) - \sum_{x \in X, y \in Y} P(x,y) \log P(x) \\
&= -H(X|Y) - \sum_{x \in X, y \in Y} P(y|x)P(x) \log P(x) \\
&= -H(X|Y) - \sum_{x \in X} P(x) \log P(x) \sum_{y \in Y} P(y|x) \\
&= -H(X|Y) + H(X),
\end{aligned} \tag{A.4}$$

where we have used the fact that $H(X) = -\sum_{x \in X} P(x) \log P(x)$, and $\sum_{y \in Y} P(y|x) = 1$.

# B   Maximum entropy distributions

Consider a neuron that can produce a discrete set of discriminable responses. The entropy of the response distribution is:

$$H = -\sum_i p_i \log(p_i),$$ (B.1)

subject to the constraint $\sum_i p_i = 1$.

To find the entropy-maximizing distribution of responses, we can define the Lagrangian $\mathcal{L}$ with Lagrange multiplier $\lambda_0$:

$$\mathcal{L}(p, \lambda_0) = -\sum_i p_i \log(p_i) + \lambda_0\left(\sum_i p_i - 1\right).$$ (B.2)

Maximizing this with respect to $p_i$ gives:

$$\frac{\partial \mathcal{L}(p, \lambda_0)}{\partial p} = 0 \implies -\log(p_i) - 1 + \lambda_0$$
$$\implies p_i = \exp(\lambda_0 - 1).$$ (B.3)

Imposing the constraint $\sum_i p_i = 1$ means:

$$\sum \exp(\lambda_0 - 1) = 1$$
$$\implies N \exp(\lambda_0 - 1) = 1$$
$$\implies p_i = 1/N.$$ (B.4)

Intuitively, given a constraint on the number of discriminable response levels, the maximum entropy distribution is flat (and corresponds to 'histogram equalization').

If we now consider a continuous distribution of responses but impose a constraint on the mean firing rate *mu*, this gives:

$$\mathcal{L}(p, \lambda_0, \lambda_1) = -\int p(r)\log(p(r))dr + \lambda_0\left(\int p(r)dr - 1\right) + \lambda_1\left(\int r p(r)dr - \mu\right),$$ (B.5)

and

$$\frac{\partial \mathcal{L}}{\partial p} = -\log(p_i) - 1 + \lambda_0 + \lambda_1 r = 0,$$
$$\implies p(r) = \exp(\lambda_0 - 1)\exp(\lambda_1 r).$$ (B.6)

Plugging this back into the equations for our constraints, we have:

$$\lambda_1 = -1/\mu,$$
$$\exp(\lambda_0 - 1) = 1/\mu,$$

which gives

$$p(r) = \frac{1}{\mu}\exp\left(-\frac{r}{\mu}\right).$$ (B.7)

Thus, given a constraint on the mean firing rate, the distribution of neural responses is exponential. It is straightforward to use the same logic to show that an additional constraint on the variance of neural responses leads to a Gaussian distribution.

# C   Bayesian ideal observer model in a switching environment

Here, we derive the ideal observer model for the scenario described in Section 2.2, which must infer the current state $\theta_t$ of the environment given the history of past sensory signals $s\tau \leq t$.

Given random variables $A$ and $B$, Bayes Rule states:

$$P(A|B) = \frac{P(B|A)P(A)}{P(B)}. \tag{C.1}$$

For three variables $A, B, C$, this becomes:

$$P(A|B, C) = \frac{P(B|A, C)P(A|C)}{P(B|C)}. \tag{C.2}$$

We can now make the substitutions: $A = \theta_t$, $B = s_t$, and $C = s_{\tau<t}$:

$$P(\theta_t|s_t, s_{\tau<t}) = \frac{P(s_t|\theta_t, s_{\tau<t})P(\theta_t|s_{\tau<t})}{P(s_t|s_{\tau<t})}. \tag{C.3}$$

This distribution must be normalized, which gives:

$$\sum_{\theta_t} P(\theta_t|s_t, s_{\tau<t}) = 1 = \sum_{\theta_t} \frac{P(s_t|\theta_t, s_{\tau<t})P(\theta_t|s_{\tau<t})}{P(s_t|s_{\tau<t})}. \tag{C.4}$$

Since the distribution in the denominator does not depend on $\theta_t$, we can pull it out of the sum and define this to be the normalization constant $\Omega$:

$$\Omega \equiv P(s_t|s_{\tau<t}) = \sum_{\theta_t} P(s_t|\theta_t, s_{\tau<t})P(\theta_t|s_{\tau<t}). \tag{C.5}$$

We now have:

$$P(\theta_t|s_t, s_{\tau<t}) = \frac{1}{\Omega}P(s_t|\theta_t, s_{\tau<t})P(\theta_t|s_{\tau<t}). \tag{C.6}$$

The second distribution can be expanded and simplified as follows:

$$P(\theta_t|s_{\tau<t}) = \sum_{\theta_{t-1}} P(\theta_t|\theta_{t-1}, s_{\tau<t})P(\theta_{t-1}|s_{\tau<t}) \tag{C.7}$$

$$= \sum_{\theta_{t-1}} P(\theta_t|\theta_{t-1})P(\theta_{t-1}|s_{t-1}, s_{\tau<t-1}). \tag{C.8}$$

Putting this together, we have:

$$P(\theta_t|s_t, s_{\tau<t}) = \frac{1}{\Omega}P(\theta_t|s_t)\sum_{\theta_{t-1}} P(\theta_t|\theta_{t-1})P(\theta_{t-1}|s_{t-1}, s_{\tau<t-1}). \tag{C.9}$$

For the specific case of a two-state environment, we can define $P_t^L \equiv P(\theta_t = \theta^L|s_t, s_{\tau<t})$, and $P_t^H = (1 - P_t^L)$. The posterior distribution is thus fully specified by a single number: the probability that the environment is in the low state, given the history of past observations. In matrix notation, we now have:

$$\begin{bmatrix} P_t^L \\ 1 - P_t^L \end{bmatrix} = \frac{1}{\Omega} \begin{bmatrix} P(s_t|\theta_t = \theta^L) & P(s_t|\theta_t = \theta^H) \end{bmatrix} \begin{bmatrix} 1 - p_s & p_s \\ p_s & 1 - p_s \end{bmatrix} \begin{bmatrix} P_{t-1}^L \\ 1 - P_{t-1}^L \end{bmatrix}. \tag{C.10}$$

Or equivalently,

$$P_t^L = \frac{1}{\Omega}P(s_t|\theta_t = \theta^L)\left[(1 - p_s)P_{t-1}^L + p_s(1 - P_{t-1}^L)\right]. \tag{C.11}$$

# D   Primer on reinforcement learning

This is meant to provide a quick-and-dirty tutorial on reinforcement learning; for a more thorough overview and background, please refer to Sutton & Barto [40], which provided the source for these notes. Massimo Vergassola covered this topic in depth in the same 2023 Les Houches Summer School; his lectures notes (in preparation) will also provide a more thorough background.

Reinforcement learning considers the process of learning to select appropriate actions $a$ in particular states $\theta$ of the environment, based on feedback from the environment in the form of rewards $r$. Note that it is typical to denote states with the variable $s$, instead of $\theta$; here we use $\theta$ to highlight that these states are typically hidden from the agent must be inferred from sensory signals, which we denote $s$. In Section 2.2, we used such latent states to parameterize distributions of sensory stimuli; in Section 2.3, the latent state specified the reward probability of different levers that could be sampled by an agent. And in Section 3, the latent state specified the animal's heading in the environment, which must be inferred from visual and self-motion signals.

The goal of a reinforcement learning agent is to maximize long-term accumulated reward, which is called the return $G_t$ and is defined as the total sum of all rewards received at future times $t' > t$:

$$G_t \equiv R_{t+1} + R_{t+2} + R_{t+3} + \ldots \tag{D.1}$$

As written, this expression assumes that all future rewards are equally important. To account for the fact the immediate rewards might be more valuable than rewards in the distant future, we can "discount" future rewards with a discount factor $\gamma \in [0,1]$:

$$
\begin{aligned}
G_t &= R_{t+1} + \gamma R_{t+2} + \gamma^2 R_{t+3} + \ldots \\
&= \sum_{t'=t+1}^{\infty} \gamma^{t'-1} R_{t'} \\
&= R_{t+1} + \gamma G_{t+1},
\end{aligned}
\tag{D.2}
$$

where $\gamma = 1$ defines a far-sighted agent that values all rewards equally, and $\gamma = 0$ defines a purely myopic agent that only values immediate rewards.

The return $G_t$ represents the sum of actual future rewards, which is not knowable to any agent. Instead, an agent can compute the *expected* future rewards, starting in state $\theta$ and following a policy $\pi(a|\theta)$. The policy specifies the probability of selecting a given action $a$ from a state $\theta$. If it is helpful to have a concrete example in mind, consider a scenario in which an agent navigates in 2D environment to gather rewards that are given when the agent reaches specific locations in the environment. A typical way to treat this scenario is to discretize the environment into a grid of locations; these locations then serve as our states (so, in a $10 \times 10$ grid, there are 100 states). At any grid location, it is often assumed that there are at most 5 available actions: stay, move 1 step up, move 1 step down, move 1 step left, or move 1 step right. A policy can then be specified as the probability of taking each of the 5 actions from each of the 100 states in the environment.

With this example in mind, we can start by computing the expected future rewards that will be obtained one timestep in the future:

$$\mathbb{E}_\pi[R_{t+1}|\theta_t = \theta] = \sum_a \pi(a|\theta) \sum_{\theta'} p(\theta'|\theta, a) r(\theta, a, \theta'), \tag{D.3}$$

where $p(\theta'|\theta, a)$ governs the dynamics of the environment and specifies the probability that the agent transitions to a state $\theta'$ when beginning in state $\theta$ and taking action $a$; the reward

function $r(\theta', a, \theta)$ specifies the reward received under the same transition. We can use Eq. D.3 to define a state-value function $v_\pi(\theta)$ that defines how 'good' it is (in terms of expected return) to begin in state $\theta$ and follow the policy $\pi$:

$$v_\pi(\theta) \equiv \mathbb{E}_\pi[G_t|\theta_t = \theta]. \tag{D.4}$$

Using the result from Eqs. D.2-D.3, we can write this as:

$$
\begin{aligned}
v_\pi(\theta) &= \mathbb{E}_\pi[R_{t+1} + \gamma G_{t+1}|\theta_t = \theta] \\
&= \sum_a \pi(a|\theta) \sum_{\theta'} p(\theta'|\theta, a)\Big[r(\theta, a, \theta') + \gamma \mathbb{E}_\pi[G_{t+1}|\theta_{t+1} = \theta']\Big] \\
&= \sum_a \pi(a|\theta) \sum_{\theta'} p(\theta'|\theta, a)\Big[r(\theta, a, \theta') + \gamma v_\pi(\theta')\Big].
\end{aligned}
\tag{D.5}
$$

This is the recursive Bellman equation for the state-value function [107]. We can also use this to define a state-action value function $q_\pi(\theta, a)$ that defines the value of starting in a state $\theta$, taking an action $a$, and following the policy $\pi$ from then onward:

$$q_\pi(\theta, a) \equiv \sum_{\theta'} p(\theta'|\theta, a)\Big[r(\theta, a, \theta') + \gamma v_\pi(\theta')\Big], \tag{D.6}$$

for which there is an analogous Bellman equation:

$$q_\pi(\theta, a) = \sum_{\theta'} p(\theta'|\theta, a)\Big[r(\theta, a, \theta') + \gamma \sum_{a'} \pi(a'|\theta')q_\pi(\theta', a')\Big]. \tag{D.7}$$

The optimal value function satisfies the Bellman optimality equation:

$$
\begin{aligned}
v_*(\theta) &= \max_a \mathbb{E}_\pi[R_{t+1} + \gamma v^*(\theta_{t+1})|\theta_t = \theta, a_t = a] \\
&= \max_a \sum_{\theta'} p(\theta'|\theta, a)[r(\theta, a, \theta') + \gamma v_*(\theta')],
\end{aligned}
\tag{D.8}
$$

and analogously for $q_*(\theta, a)$.

Note that one can use a value function $v_\pi(\theta)$ or action-value function $q_\pi(\theta, a)$ to define a policy. For example, a purely greedy policy would select actions that maximize the current value:

$$A = \text{argmax}_a q(\theta, a). \tag{D.9}$$

If $q = q_*$ is the optimal action-value function, then Eq. D.9 defines the optimal policy $\pi_*(a|\theta) = \text{argmax}_a(q_*(\theta, a))$. However, if $q$ is not optimal, it can be advantageous to use a policy that balances exploitative actions (which maximize the current estimate of $q$) with exploratory actions (which have lower expected value but could result in higher long-term payoffs). A simple version of such a policy is called an 'epsilon-greedy' policy, in which the agent chooses the greedy (exploitative) action with probability $(1-\epsilon)$, and chooses a random (exploratory) action with probability $\epsilon$, where $\epsilon \in [0, 1]$ is typically chosen to be small. Alternatively, one can tie the degree of exploration to the value function by choosing a 'softmax' policy:

$$\pi(a|\theta) \propto \exp(\beta q(\theta, a)), \tag{D.10}$$

where $\beta \to \infty$ drives purely exploitative actions that maximize $q$, and $\beta \to 0$ drives purely random actions.

A goal of reinforcement learning is to use value functions to determine the optimal policies that maximize the return $G$. Historically, there have been three main sets of methods for

learning value functions. The first are dynamic programming methods, which assume a model of the environment in the form of $p(\theta'|\theta, a)$ and $r(\theta, a, \theta')$, and learn $v_*$, $q_*$, and $\pi_*$ via bootstrapping (i.e., updating an estimate of one quantity based on an estimate of another quantity). Value iteration, which we introduced in Section 2.3, is one such method that converts Eq. D.8 into an update rule for the value function:

$$v_{t+1}(\theta) = \max_a \sum_{\theta', r} p(\theta'|\theta, a) \Big[ r(\theta, a, \theta') + \gamma v_t(\theta') \Big]. \tag{D.11}$$

The second set of methods are Monte Carlo methods, which do not assume a model of the environment, and instead directly estimate $v^*$, $q^*$, and $\pi^*$ via sampling (i.e., from the outcome of simulated experiences). An example of a Monte Carlo update is the following:

$$v_{t+1}(\theta) = v_t(\theta) + \alpha[G_t - v_t(\theta)], \tag{D.12}$$

where the return $G_t$ must be computed from the outcome of simulated experiences. $G_t$ is then used as a target to update $v_t$.

A third class of methods, temporal difference (TD) methods, combine aspects of dynamic programming and Monte Carlo methods to learn from experience via bootstrapping [40]. The simplest TD method replaces the return $G_t$ in Eq. D.12 with the estimate $G_t \approx R_{t+1} + \gamma v_t(\theta_{t+1})$:

$$v_{t+1}(\theta_t) = v_t(\theta_t) + \alpha\big[R_{t+1} + \gamma v_t(\theta_{t+1}) - v_t(\theta_t)\big], \tag{D.13}$$

$$q_{t+1}(a_t, \theta_t) = q_t(a, \theta_t) + \alpha\big[R_{t+1} + \gamma q_t(a_{t+1}, \theta_{t+1}) - q_t(a_t, \theta_t)\big]. \tag{D.14}$$

Here, $\delta_t = R_{t+1} + \gamma v_t(\theta_{t+1}) - v_t(\theta_t)$ (and similarly $\delta_t = R_{t+1} + \gamma q_t(a_{t+1}, \theta_{t+1}) - q_t(a_t, \theta_t)$) is often referred as the TD error. Eq. D.14 forms the basis of the so-called SARSA learning algorithm. Note that the updating of the value functions depends on the actions taken and rewards received, which in turn depend on the policy used to select those actions. Thus, when using one of the policies defined in Eqs. D.9-D.10, the actions selected via the policy will change over time as the value functions are updated via Eqs. D.13-D.14.

With an algorithm like SARSA, the agent can only update the value of states as these states are visited. To update states that were visited in the past, we can augment this update rule with an eligibility trace $Z_t(\theta)$ that allows previously-visited to be eligible for updating:

$$v_{t+1}(\theta_t) = v_t(\theta_t) + \alpha\big[R_{t+1} + \gamma v_t(\theta_{t+1}) - v_t(\theta_t)\big]Z_t(\theta_t), \tag{D.15}$$

where $Z_t(\theta)$ follows its own update rule:

$$Z_t(\theta) = \begin{cases} \lambda\gamma Z_{t-1}(\theta), & \theta \neq \theta_t, \\ 1 + \lambda\gamma Z_{t-1}(\theta), & \theta = \theta_t. \end{cases} \tag{D.16}$$

Here, $\lambda \in [0, 1]$ is a trace-decay parameter that specifies how quickly the eligibility of each state will decay over time. When $\lambda = 0$, only the current state can be updated; when $\lambda = 1$, the eligibility of an unvisited state falls by $\gamma$ with each timestep. Larger values of $\lambda$ cause the agent to associate previously-visited states with current rewards.

# E  Derivation of policy gradient algorithm

This follows the derivation given in Sutton & Barto [40]. We begin by defining a performance measure $J(\vec{\psi})$ that we want to maximize, given a policy $\pi(a|\theta; \vec{\psi})$ parameterized by $\vec{\psi}$:

$$J(\vec{\psi}) \equiv v_{\pi_{\vec{\psi}}}(\theta), \tag{E.1}$$

where $v_{\pi_{\vec{\psi}}}(\theta)$ is the value for a policy $\pi$, parameterized by $\vec{\psi}$, starting in state $\theta$.

We want to find the parameter update that increases performance over time:

$$
\begin{aligned}
\nabla_{\vec{\psi}} v_\pi(\theta) &= \nabla_{\vec{\psi}} \left[ \sum_a \pi(a|\theta) q_\pi(\theta, a) \right] \\
&= \sum_a \left[ q_\pi(\theta, a) \nabla_{\vec{\psi}} \pi(a|\theta) + \pi(a|\theta) \nabla_{\vec{\psi}} q_\pi(\theta, a) \right].
\end{aligned}
\tag{E.2}
$$

The second term in this sum, *, can be written as:

$$
\begin{aligned}
* &= \pi(a|\theta) \nabla_{\vec{\psi}} \sum_{\theta', r} p(\theta', r|\theta, a)(r + v_\pi(\theta')) \\
&= \pi(a|\theta) \nabla_{\vec{\psi}} \sum_{\theta'} p(\theta'|\theta, a) \nabla_{\vec{\psi}} v_\pi(\theta').
\end{aligned}
\tag{E.3}
$$

We can now note that

$$
\nabla_{\vec{\psi}} v_\pi(\theta) = \sum_a \left[ q_\pi(\theta, a) \nabla_{\vec{\psi}} \pi(a|\theta) + \pi(a|\theta) \sum_{\theta'} p(\theta'|\theta, a) \nabla_{\vec{\psi}} v_\pi(\theta') \right],
\tag{E.4}
$$

and in a similar manner, $\nabla_{\vec{\psi}} v_\pi(\theta')$ can be 'rolled out' and expressed in terms of $\nabla_{\vec{\psi}} v_\pi(\theta'')$:

$$
\begin{aligned}
\nabla_{\vec{\psi}} v_\pi(\theta) = \sum_a \Bigg[ & q_\pi(\theta, a) \nabla_{\vec{\psi}} \pi(a|\theta) + \pi(a|\theta) \sum_{\theta'} p(\theta'|\theta, a) \\
& \times \sum_{a'} \left[ q_\pi(\theta', a') \nabla_{\vec{\psi}} \pi(a'|\theta') + \pi(a'|\theta') \sum_{\theta''} p(\theta''|\theta', a') \nabla_{\vec{\psi}} v_\pi(\theta'') \right] \Bigg]..
\end{aligned}
\tag{E.5}
$$

Note that the first two rows of the above equation are identical in form, but are weighted by the probability of transitioning between states under policy $\pi$. Replacing $\theta$ with its value sampled at time $t$, we can write:

$$
\begin{aligned}
\nabla_{\vec{\psi}} J(\vec{\psi}) &= \mathbb{E}_\pi \left[ \sum_a q_\pi(\theta_t, a) \nabla_{\vec{\psi}} \pi(a|\theta_t; \vec{\psi}) \right] \\
&= \mathbb{E}_\pi \left[ \sum_a \pi(a|\theta_t; \vec{\psi}) q_\pi(\theta_t, a) \frac{\nabla_{\vec{\psi}} \pi(a|\theta_t; \vec{\psi})}{\pi(a|\theta_t; \vec{\psi})} \right] \\
&= \mathbb{E}_\pi \left[ \sum_a \pi(a|\theta_t; \vec{\psi}) q_\pi(\theta_t, a) \nabla_{\vec{\psi}} \log \pi(a|\theta_t; \vec{\psi}) \right],
\end{aligned}
\tag{E.6}
$$

where we have multiplied and divided by $\pi(a|\theta_t, \vec{\psi})$ in the second line, and written $\nabla x / x = \nabla \log x$ in the third line. If we now replace $a$ by a sampled action $a_t \sim \pi(\cdot|\theta_t)$, we have:

$$
\nabla_{\vec{\psi}} J(\vec{\psi}) = \mathbb{E}_\pi \left[ G_t \nabla_{\vec{\psi}} \log \pi(a_t|\theta_t; \vec{\psi}) \right],
\tag{E.7}
$$

where $G_t$ is the return at time $t$, and where we have written $q_\pi(\theta_t, a_t) = \mathbb{E}_\pi[G_t|\theta_t, a_t]$. The argument of this expectation can be sampled on every timestep, and the expectation of this quantity is equal to the gradient $\nabla_{\vec{\psi}} J(\vec{\psi})$ [108]. This gives the update rule that we used in Eq. 37:

$$
\vec{\psi}_{t+1} = \vec{\gamma}_t + \alpha R_t \nabla_{\vec{\psi}} \log \pi(a_t|\theta_t; \vec{\psi}),
\tag{E.8}
$$

where we replaced the full return $G_t$ with the instantaneous reward $R_t$ (i.e., we considered a purely myopic agent).

## F   Structuring a motor drive about a goal heading

We consider a motor drive $m$ of the form:

$$m = \int_0^{2\pi} d\theta \, \cos(\theta - \theta_c) f_g(\theta),$$

(F.1)

where $\cos(\theta - \theta_c)$ is an activity profile that encodes the current compass heading $\theta_c$, and where $f_g(\theta)$ is an arbitrary activity profile that encodes the goal heading.

Expanding the cos, we have:

$$
\begin{aligned}
m &= \int_0^{2\pi} d\theta \, \frac{1}{2} \left( e^{i(\theta - \theta_c)} + e^{i(\theta - \theta_c)} \right) f_g(\theta) \\
&= \frac{e^{-i\theta_c}}{2} \int_0^{2\pi} d\theta \, e^{i\theta} f_g(\theta) + \frac{e^{i\theta_c}}{2} \int_0^{2\pi} d\theta \, e^{-i\theta} f_g(\theta),
\end{aligned}
$$

(F.2)

where

$$\int_0^{2\pi} d\theta \, e^{\pm i\theta} f_g(\theta) \equiv r_g e^{\pm i\theta_g},$$

(F.3)

defines the circular mean of the function $f_g(\theta)$, and $r_g$ and $\theta_g$ are the modulus and orientation of the circular mean, respectively.

This allows us to write:

$$
\begin{aligned}
m = m(\theta_c, \theta_g) &= \frac{r_g}{2} \left( e^{-i\theta_c} e^{i\theta_g} + e^{i\theta_c} e^{-i\theta_g} \right) \\
&= r_g \cos(\theta_c - \theta_g)..
\end{aligned}
$$

(F.4)

Thus, when integrated over a full period, the product of the goal and compass activity profiles will be largest when the compass heading is aligned with the circular mean of the goal activity profile. We can then use this circular mean to define the goal heading.

If we now add a phase shift $\phi$ to the compass heading, this becomes:

$$
\begin{aligned}
m(\theta_c, \theta_g, \phi) &= \int_0^{2\pi} d\theta \, \cos\left(\theta - (\theta_c + \phi)\right) f_g(\theta) \\
&= r_g \cos(\theta_c + \phi - \theta_g),
\end{aligned}
$$

(F.5)

which peaks at $\theta_c = \theta_g - \phi$. Different phase shifts can then be used to generate motor drives that peak at different compass headings relative to the goal heading. Importantly, the motor drive will retain a sinusoidal profile regardless of the shape of $f_g(\theta)$, and thus regardless of the goal heading $\theta_g$. As a result, we can use any method we like for updating the goal activity profile, and it will not disrupt the structure of the motor drive.

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
