# Peer review of "Normative approaches to neural coding and behavior"

_SciPost Physics Lecture Notes, doi:SciPost Phys. Lect. Notes 83 (2024)_

## Round 1 · Referee Report · Anonymous (Referee 1) · 2024-2-29

Strengths

(1) Great clarity
(2) Many useful examples
(3) Well structured
(4) Highly pedagogical

Weaknesses

Only a few typos

Report

The lecture notes are well written and provide an ideal introduction to the field for PhD students in physics and biology. The presentation provides a very clear overview on the three main approaches under consideration (efficient coding, inference, action selection) that are introduced via useful examples and fruitfully compared.

Requested changes

I could only find some small typos:

  • Line 44: "difference timescales" --> "different"
  • Line 103: "they systems" --> "these"
  • Line 212: "as an one" --> "as one"
  • Line 310, Eq. (7): $s$ instead of $s_t$ in the denominator
  • Line 341, Eq. (10): the sum should run over $\theta_t$
  • Line 348: why is there an arrow over the standard deviation?
  • Line 400: "in turn impacts" --> "impact"
  • Line 404: "R,L"--> "H,L"
  • Line 508: class "of" behaviors
  • Line 796: son "of" Werner

Minor comments: - Line 280: It is clear from the context, but maybe in addition to point (4) one could say that the underlying assumption is that the stimulus distribution is known. - Line 301, Eq. (6): the reader may wonder why we take the MSE instead of the maximum a posteriori given that we know there are only two options for $\theta$.

---

## Round 1 · Referee Report · Anonymous (Referee 2) · 2024-4-6

Strengths

Great paper. Few small typos.

Weaknesses

Eq 2 -- there's some issue with definition here, as when H>1, redundancy becomes negative. While it's possible to have a negative redundancy, H=1 shouldn't an arbitrary entropy value, at which redundancy changes sign.

When discussing RL in Sec 2.3, it'd probably be useful to refer to the chapter by Vergassola, where RL will be discussed in detail (and a similar reference from Vergassola would be needed, too).

Not crucial, but in 3.1, when discussing local excitation-global inhibition, it might be worthwhile, for pedagogical purposes, to connect this to Turing patterns and to the LEGI model in chemotaxis. Maybe more relevant, some reference to balanced networks might also be useful.

In discussion round Fig 17, how crucial is the form of ReLU nonlinearity? For a different nonlinearity, is it always possible to select J_E, such that the distinct orientations form a Goldstone mode, rather than a collection of small bumps and valleys?

Report

yes, criteria met, publish

---

## Round 1 · Referee Report · Anonymous (Referee 3) · 2024-4-9

Strengths

  1. Pedagogical presentation of normative approaches in neural coding.
  2. Clear presentation of the fly's navigation system.
  3. Beautiful figures and illustrations.

Weaknesses

none

Report

This paper summarizes the lecture given by the author at the Les Houches summer school. The article clearly summarizes the content of the lecture and reviews important results in the field in a pedagogical way ideally suited for students and researchers interested in learning the topic.

Recommendation

Publish (surpasses expectations and criteria for this Journal; among top 10%)

---

## Editorial Decision

published